# Absolute protein quantification using fluorescence measurements with FPCountR

**Eszter Csibra** [1] ✉ **& Guy-Bart Stan** [1] ✉

This paper presents a generalisable method for the calibration of fluorescence readings on microplate readers, in order to convert arbitrary fluorescence units into absolute units. FPCountR relies on the generation of bespoke fluorescent protein (FP) calibrants, assays to determine protein concentration and activity, and a corresponding analytical workflow. We systematically characterise the assay protocols for accuracy, sensitivity and simplicity, and describe an 'ECmax' assay that outperforms the others and even enables accurate calibration without requiring the purification of FPs. To obtain cellular protein concentrations, we consider methods for the conversion of optical density to either cell counts or alternatively to cell volumes, as well as examining how cells can interfere with protein counting via fluorescence quenching, which we quantify and correct for the first time. Calibration across different instruments, disparate filter sets and mismatched gains is demonstrated to yield equivalent results. It also reveals that mCherry absorption at 600 nm does not confound cell density measurements unless expressed to over 100,000 proteins per cell. FPCountR is presented as pair of open access tools (protocol and R package) to enable the community to use this method, and ultimately to facilitate the quantitative characterisation of synthetic microbial circuits.

There is a growing awareness that tackling the challenge of synthetic circuit design requires the synthesis of empirical characterisation data on genetic parts with mathematical modelling approaches for predicting and realising desired behaviours[1–3]. However, there are numerous challenges in integrating experimental data with quantitative frameworks, as experimental data is typically acquired in relative or arbitrary units (specific to instruments and their respective settings), which cannot be converted into useful units and therefore limits our ability to make comparisons between experiments and models.

Fluorescent proteins (FPs) are our most versatile tools for the assessment of synthetic genetic element performance. Since their discovery, FPs have rightly been recognised as uniquely valuable reporter proteins for quantitative characterisation[4,5], since they do not require the addition of exogenous components to fluoresce. This makes their use easy and cost effective. Various laboratory instruments allow the characterisation of fluorescent systems in a wide range of dimensions and scales—through direct visualisation (using fluorescence microscopy), via single-cell fluorescence analysis (using flow cytometry), or via timecourse kinetic data acquisition (using microplate readers).

The 'protein quantification problem' of reporting GFP levels acquired by such instruments in 'relative fluorescent units' (RFU) has been widely recognised in synthetic biology[6]. This recognition has led to the adoption of calibration standards such as fluorescein, a small molecular fluorophore with similar excitation and emission characteristics to GFP. Fluorescein can be used to calibrate a given instrument by converting the instrument's arbitrary RFU output into units of 'molecules of equivalent fluorescein' (MEFL). This technique has been demonstrated to enable the comparison of GFP expression data gathered from different instruments as well as across laboratories[6,7].

[1]Department of Bioengineering, Imperial College Centre for Synthetic Biology (IC-CSynB), Imperial College London, London SW7 2AY, UK.
✉e-mail: e.csibra@imperial.ac.uk; g.stan@imperial.ac.uk

While it is now approaching mainstream usage in synthetic biology, the conversion of green fluorescence values into MEFL is arguably not the most important type of quantification required for building synthetic circuits. Three aspects of the protein quantification problem remain elusive. First, fluorescein is only a good calibrant for green fluorescent proteins, leaving blue, yellow, orange and red FPs uncalibrated. Second, fluorescein can only provide a conversion to units of fluorescein, whereas what is actually needed is a conversion to units of protein. Currently, most experiments cannot even reveal the order of magnitude at which FPs are being expressed (i.e. 10 vs 100,000 molecules per cell)–in contrast to what is possible with RNA sequencing[8]. Third, while fluorescein can allow the comparison of GFP levels between instruments and laboratories, it cannot address the comparison between two different FPs in the same circuit. This is only possible if RFUs from both FPs can be separately converted into molecular units of protein. Some have attempted to tackle this by attempting to predict the relative brightness of FPs using theoretical values[9,10], but such calibrations make a number of assumptions, for instance about translation rate equivalence across constructs, that require validation before they can be adopted.

Fortunately, there is a reasonably simple solution. The ideal calibrant in molecular biology is considered to be a purified sample of the molecule to be measured–in this case, the fluorescent protein itself. While purified FPs are not generally commercially available, they can be produced 'in-house', thereby providing that crucial direct link between relative fluorescence units and molecules of protein. Indeed, FP calibration has been proposed in the past, though its use has been limited to microscopy[11–13] and remains rare for microplate assays[14,15]. We suspect this is due to (a) an underappreciation that absolute quantification is possible without 'omics', (b) a reticence to try unfamiliar biochemical protocols that are not usually part of the synthetic biology or microbiology repertoire, and (c) doubt that such protocols could be accurate or sensitive enough for general usage. These motivated us to develop a general, yet simple-to-use calibration protocol of this nature.

In what follows, we outline our optimised calibration method and present it as a pair of resources: a wet lab protocol, called FPCount (available on protocols.io) and an accompanying analysis package, called FPCountR (available on GitHub). We present data showing the development of this protocol, and systematically characterise the biochemical and analytical requirements of an accurate and sensitive calibration. We also present an absorbance-based fluorescent protein assay, which acts both to simplify the method to remove the requirement for protein purification, and to make it more sensitive and robust. Using FPCountR, we show that conversion to molecular units can be used to calibrate across different instruments, disparate filter sets and mismatched gains to yield equivalent results. We demonstrate that conversion to absolute units allows the user of our method to compare the protein production efficiency of different fluorescent proteins expressed from an otherwise identical vector in molecules per cell, or as a molar concentration. Finally, we demonstrate that this method can be used to quantitatively evaluate the experimental protocols themselves, such as the extent to which red fluorescent proteins confound optical density readings in timecourse assays.

## Results

Our aim for this work was to develop a generalisable method for FP calibration that could be used by any group wishing to calibrate fluorescence readings on microplate readers to molecular units. To do this, we defined a number of key aims for our proposed method. First, it should be accurate and sensitive, as we need the method to correctly estimate molecule numbers within cells, and as protein yields from small-scale purifications are typically modest. Second, the calibration protocol should be as simple as possible and adapted ideally such that each respective assay may be carried out in 96-microwell plate format

using the same plate reader that is being calibrated. This way, multiple fluorescent proteins may be calibrated at once, and end users do not require any additional instrumentation. Third, the method should be suitable for the particular characteristics of fluorescent proteins. These proteins are smaller and structurally distinct from typical protein calibrants such as bovine serum albumin (BSA), and are known to present certain challenges for quantification due to light absorption by their chromophores. Thus, any assay developed for non-fluorescent proteins requires a separate validation on FPs to demonstrate that they are also adequate for this class of proteins. Finally, we wanted to enable the easy analysis of the data, by (i) enabling easy conversion of raw calibration data into a conversion factor that links the arbitrary fluorescence output of a protein with its quantity in molecular units, and (ii) allowing easy conversion of data from all future timecourse data from that instrument to produce outputs in (e.g. GFP) molecules, rather than relative (e.g. green) fluorescence units. An overview of the FPCount fluorescent protein calibration protocol is illustrated in Fig. 1.

### A purification protocol for obtaining protein calibrants

In order to obtain our fluorescent protein calibrants, they must first be produced by overexpression, and purified. Protein purification methodologies can often be highly complex, requiring specialist expertise and instrumentation. To alleviate this, our protocol was explicitly designed to be as straightforward as possible, involving only the minimum number of required steps, and using commonly available reagents. It was also designed to be amenable to small-volume purifications to enable the calibration of multiple proteins in parallel. The protocol is summarised in Fig. 2a. A standardised FP expression vector was constructed from an arabinose-inducible His-tagged FP construct in a high-copy Standardised European Vector Architecture (SEVA) vector (Fig. 2b). The use of high-copy vectors and overnight expression was designed to maximise protein production, and the temperature was dropped to 30 °C to minimise misfolding. Cells were lysed using sonication to avoid the requirement to add chemical components that may interfere with downstream processes, such as EDTA (with His-tag purification), detergents (with protein quantification), or unknown components of commercial lysis reagents. Insoluble proteins were removed via centrifugation and sodium dodecyl sulphate polyacrylamide gel electrophoresis (SDS-PAGE) was used to confirm that the majority of the expressed FP was in the soluble fraction (Fig. 2c). Proteins were purified using His-tag affinity purification, as His tags are small in size, making them unlikely to compromise fusion protein function. Cobalt resin was used as the affinity matrix as it has higher specificity for His tags than nickel resin, and was therefore expected to co-isolate fewer impurities. The quality of purified FP calibrants was verified by SDS-PAGE and fluorescence excitation and emission scanning (Fig. 2d, e, Supplementary Fig. 1). Purified calibrants were of consistently good purity and yield.

### Conducting a plate reader calibration

The calibration of plate readers with fluorescein has traditionally been conducted using a dilution series of known concentrations of fluorescein, subjected to a fluorescence assay in the plate reader whose calibration is desired (measurement of relative fluorescence units, RFU). The results are used to relate fluorescein molecule number to RFU to obtain a conversion factor, which can in turn be used to convert RFU readouts from experimental data into MEFL units[6]. For protein calibrants, one additional step is required: protein concentration determination.

In our initial protocols, we opted for the bicinchoninic acid (BCA) assay due to its sensitivity, ease of use and low protein-to-protein variability[16]. In addition, microplate-optimised reagents for a 'micro-BCA' were available from ThermoFisher, with excellent reported sensitivities (to 2 ng/μl). Pilot tests showed an inhibitory effect of the Tris and NaCl in the elution buffer (Supplementary Fig. 2A) suggesting that

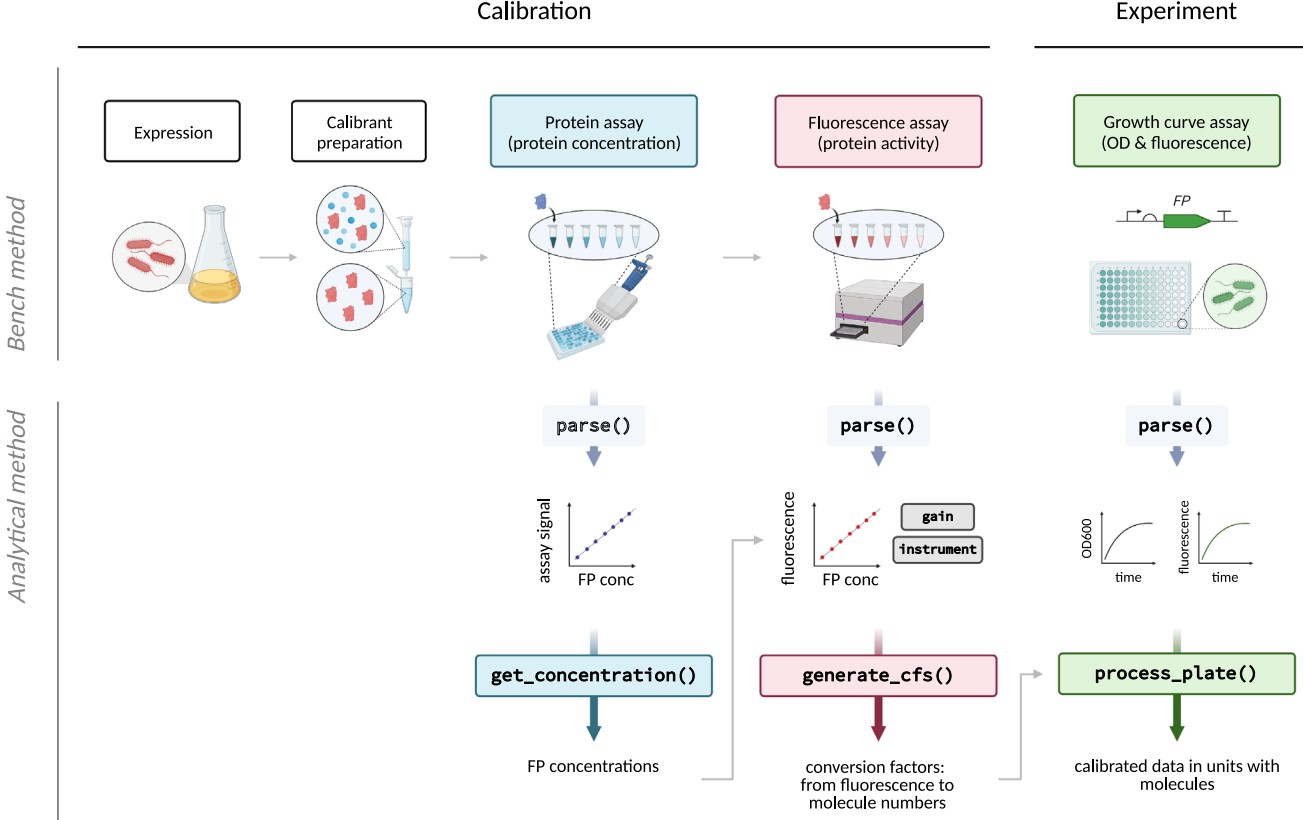

**Fig. 1 | Overview of fluorescent protein calibration workflow using FPCountR.** A calibration workflow is described (left), followed by a demonstration of how this calibration can be used to convert experimental data from arbitrary fluorescence units per optical density into molecules per cell (right). The calibration workflow consists of a wet lab protocol (top, available on protocols.io) and an analysis package (bottom, available on GitHub). In brief, the protocol describes how to prepare fluorescent protein calibrants by expression and purification, though the latter step is optional as lysates allow accurate calibration without the need for purification. The protocol also describes how to collect data for the calibration for both the protein assay to determine protein concentration, as well as the fluorescence assay to determine protein activity. The analytical workflow is provided as an open-source R package, complete with functions that enable the extraction of protein concentrations from protein assay data, conversion factors (arbitrary fluorescence units per molecule) from a combination of protein and fluorescence assay data, and functions that allow users to convert experimental data into absolute units. Figure created with Biorender.com.

buffer exchange would be necessary for high assay sensitivity. Normalised measurements using a BSA standard obtained with the kit were fitted to a polynomial equation to obtain a standard curve (Supplementary Fig. 2D, ii). This was then used to predict the concentrations of FPs prepared as serial dilutions by first removing values under the reported threshold of sensitivity and fitting a linear model through the rest of the values (Supplementary Fig. 2D, ii and iii). An extra step was added to the recommended protocol (Supplementary Fig. 2A, ii and Supplementary Fig. 2C) to account for baseline absorbance of red FPs in the A562 range. These calculations are handled in the FPCountR package by the get_conc_bca() function. Using the resultant predicted protein concentrations and fluorescence assay data on the same FP dilution series, an adapted version of the generate_cfs() function from flopR[7] was used to generate conversion factors (RFU/molecule) for mCherry in a Tecan Spark plate reader for the red FP-typical filter set (ex 560/20, em 620/20; Supplementary Fig. 3).

### Development of the A280 assay for FP concentration

We sought to verify the accuracy of the BCA assay by re-quantifying our FPs with a second method that is likely to give reliable concentration estimates. While a wide variety of protein assays exist, the only widely-used 'absolute' assay that does not require a calibrant is the A280 assay. As the name suggests, it quantifies protein concentration via light absorbance at 280 nm, where three amino acid residues are known to absorb light in a way that has been shown to be approximately additive[17]. This means that a reasonable prediction of light absorbance at 280 nm can be made for any pure protein of known primary sequence by way of an extinction coefficient (EC; expected light absorption for a given concentration of protein). As sample absorbance relates to molecular concentration according to Beer's law, i.e., $A = EC*C*L$ (where $A$ is the absorbance, $EC$ is the extinction coefficient ($M^{-1} cm^{-1}$), $C$ is the concentration (M), and $L$ is the path length (cm)), the protein concentration may be calculated from absorbance using only the extinction coefficient and the path length. The most common formats for A280 measurements are laborious, single-throughput cuvette- or Nanodrop-type measurements, requiring the adaptation of the standard A280 protocols for use in 96-well microplates. We have summarised the requirements for such an adaptation in Supplementary Note 1 (along with Supplementary Figs. 4–8 and Supplementary Table 3). In brief, the best results for A280 assays were obtained by using UV-clear plastic, removing additives, correcting for path length variation, and correcting for light scatter. This required the collection of an absorbance spectrum from 200 to 1000 nm rather than just one reading at 280 nm, and is processed by two consecutive FPCountR functions, plot_absorbance_spectrum() and get_conc_a280() (Supplementary Figs. 6–7).

### Systematic protocol testing allows method validation

We sought to conduct a systematic assessment of the BCA and A280 methods by testing three spectrally distinct FPs in two buffers, assessed with both assays in parallel (Fig. 3). The chosen FPs (mTagBFP2,

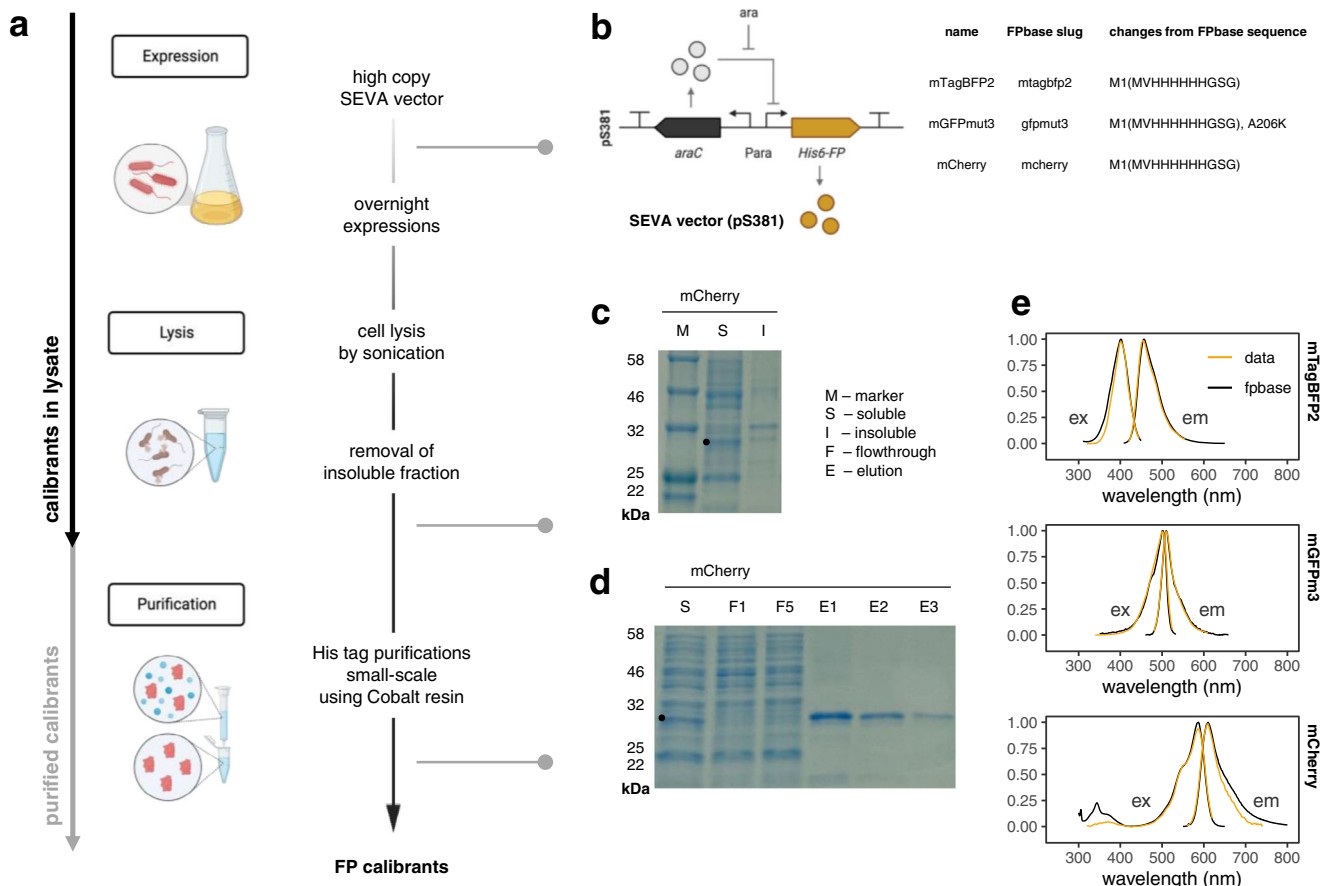

**Fig. 2 | Preparation of fluorescent protein calibrants. a** Protocol summary. The use of high-copy vectors and overnight expression was designed to maximise protein production. Cells were lysed using sonication, to avoid the requirement to add chemical components that may interfere with downstream processes. Insoluble proteins were removed via centrifugation. Proteins were purified using His-tag affinity purification with cobalt resin which was expected to co-isolate fewer impurities. The arrows on the left represent the steps required to prepare purified calibrants (grey) vs. calibrants in lysate (black). **b** Vector and FP design. A standardised FP expression vector was constructed from an arabinose-inducible His-tagged FP construct in a high-copy SEVA vector. Three commonly used FPs from across the spectral range were chosen for testing this protocol: mTagBFP2, mGFPmut3, and mCherry. A table of the three proteins illustrates any changes in protein sequence compared to their FPbase counterparts, showing identical matches with the exception of affinity tags and a monomerising mutation for GFPmut3. **c** Expression and solubility verification. SDS-PAGE analysis of lysates after separation of the insoluble fraction was used to make sure that most of the fluorescent

protein was soluble. The displayed SDS-PAGE is from an mCherry purification, showing the separation of the soluble (S) vs. insoluble (I) fraction, next to the protein marker (M) on a 12% gel. **d** Purification verification. SDS-PAGE analysis after purification was used to confirm the success of purifications. The displayed SDS-PAGE is from an mCherry purification, showing the separation of the soluble (S) fraction, next to two flowthrough (F) fractions from the binding steps showing efficient FP binding to the cobalt resin, and three elution (E) fractions. **e** Fluorescence spectra. Fluorescence spectral scans (ex, excitation; em, emission) were used to confirm that the purified FP behaves as expected. The figure shows obtained spectra (normalised such that the highest value = 1) fitted to a loess model with a 95% confidence interval (orange) overlaid with FPbase spectra (black) for each FP. Displayed spectra represent one sample measured in duplicate, that is representative of at least 2 independently purified batches of calibrant. Figure panels **a** and **b** created with Biorender.com. Source data are provided as a Source data file.

mGFPmut3 and mCherry) are widely used, monomeric, reasonably fast-maturing and bright. All three are almost identical on the protein level to their FPbase entries (Fig. 2b; 'Methods'; Supplementary Data), with the exception that they all have a His$_6$ tag N-terminal extension, and mGFPmut3 includes a well-defined monomeric A206K mutation[18,19]. The chosen buffers (T5N15 (5 mM Tris-HCl pH 7.5, 15 mM NaCl) and T5N15 with protease inhibitors) were both compatible with the microBCA assay (Supplementary Fig. 2B), however, pilot studies suggested they might have different effects on the A280 assay. Following purification, FPs were initially dialysed to remove additives, then re-dialysed into the respective assay buffer (Fig. 3a). Each FP:buffer combination was then serially diluted, and subjected to an absorbance scan (200–1000 nm measurement, for the A280), a fluorescence assay (fluorescence measurement with appropriate filters for each FP) and a microBCA assay (reagent addition, incubation and A562 measurement). The results of this comparative test are shown in Fig. 3b, c, Supplementary Fig. 9 and Supplementary

Tables 1–3. Broadly, the results from each assay validate those of the other assay: the measured concentration of each FP using the microBCA and A280 assays are within 2-fold of each other for most samples (Fig. 3c) and apparent linear ranges reach 1 ng/µl for most dilution series (Fig. 3b). For comparison, the reported sensitivity on the Nanodrop is 100 ng/µl[20]. We observed some buffer sensitivity for both assays—the microBCA produces more linear results in the buffer containing protease inhibitors, whereas the A280 does better in the buffer without them. Overall, the A280 assay produces data that fits better to a linear regression than the microBCA assay, suggesting it may be more reliable at the relatively low concentrations used in these assays (Supplementary Table 2). Buffer effects were also apparent for the fluorescence assay (Supplementary Note 2, Supplementary Fig. 10). Conversion factors obtained from different purification batches gave similar estimates where concentration estimates were made using optimal assay:buffer pairings (Supplementary Fig. 9B).

## ECmax assay performs better than conventional assays

We trialled a third protein assay during this experiment, designated here as the 'ECmax' method. The principal idea behind this assay is that the A280 extinction coefficient is not the only known extinction coefficient for FPs. FPs also possess an extinction coefficient ('EC') corresponding to their light absorption at their peak ('max') excitation wavelength. As the 'ECmax' of most FPs is available on FPbase, we can automate its retrieval using the FPbase API[21]. (FPbase (www.fpbase.org/) is an open-source, community-editable database of fluorescent proteins and their properties. Each FP in the database contains its own page with a structured set of properties, such as its primary protein sequence, extinction coefficient and fluorescence spectra. Essentially

**Fig. 3 | Systematic characterisation reveals ECmax assay that allows FP calibration without protein purification. a–c** FP calibration using purified proteins. **a** Calibrant preparation and assay workflow. Purified FPs in elution buffer (with protease inhibitors, pi) were dialysed into T5N15, then again into T5N15±pi. Serial dilutions of each FP in both buffers were prepared, and subjected to three protein assays (microBCA, A280 and ECmax assay) as well as the fluorescence assay. **b** Measured protein concentrations using different methods and buffers. An example using mTagBFP2 (for full results see Supplementary Fig. 9). Points represent the mean of the duplicate values. Any missing data points had concentrations recorded as being below 0.01 ng/μl. **c** Assays compared to the A280. The raw data of each serial dilution, as displayed in (**b**), was fitted to a linear model and used to estimate the concentration of the first sample in the series (where the dilution factor = 1), according to methods for each assay. For a given FP batch (set), the microBCA and ECmax assay error was calculated by taking the fold difference in concentration predicted by the named assay, versus that predicted by the A280

assay using T5N15 buffer. Each point therefore represents one value for each FP batch. For the full dataset, see Supplementary Table 1. **d–f** FP calibration in crude lysates. **d** Calibrant preparation and assay workflow. Calibration in lysate requires fewer steps: affinity purification is omitted, buffer exchange is not required, and dilution series are subject to only two assays requiring no commercial reagents or incubation steps. **e** Measured protein concentrations with ECmax assay. Dilution series using lysates obtained by sonication (orange) or chemical lysis (green), were measured with an absorbance scan. The top dilution of each series using chemical lysis was removed due to excessive sample scatter. For the full dataset, see Supplementary Fig. 11A. **f** Conversion factor comparisons comparing lysates vs purified proteins. The fold difference (lysate/purified protein) conversion factor was calculated at each gain. Plots display mean fold differences across the gains and error bars represent standard deviations. For the full dataset, see Supplementary Fig. 11B. Figure panels **a** and **d** created with Biorender.com. Source data are provided as a Source data file.

all commonly used FPs have entries on this database, along with rarely used variants, and these are accessible via its API.) The analytical processing steps for the ECmax assay (in get_conc_ecmax()) are similar to those of the A280 assay (Supplementary Fig. 8), and require no further readings. As the maximal absorbance peaks for all FPs tested were higher than those at 280 nm (Supplementary Fig. 1B, Supplementary Table 4), we anticipated that the ECmax assay would be more sensitive. Further, as protease inhibitors absorb at wavelengths under 300 nm, we hypothesised this assay may be less buffer-sensitive. Compellingly, both appear to be true: we consistently found that the ECmax assay produced larger linear ranges and lower limits of detection than the other assays (approaching 0.1 ng/μl, 10-fold better than the A280 assay and 1000-fold better than a Nanodrop), and that it produced almost indistinguishable results whether or not the buffer contained protease inhibitors (Fig. 3b, c, Supplementary Fig. 9, Supplementary Table 1). In addition, predictions from the ECmax assay closely match those from the A280, typically predicting concentrations matching at 80–100% those of the expected result (rather than 170–220% for microBCA; Fig. 3c, Supplementary Table 1), suggesting an error rate of <20% compared to the reference value from the A280 assay. For these reasons, we propose that the ECmax assay would be the most robust assay to include in a simple calibration protocol and will proceed using concentrations calculated from ECmax assays in what follows.

### Robustness of calibration protocols using purified FPs

To further investigate the reproducibility of this calibration method, we completed two more independent repeats of calibrations with all three FPs using the T5N15pi buffer and the ECmax assay. From this data, we observed that one of our calibration runs obtained with mTagBFP2 in our original experiments (Supplementary Fig. 9A, mTagBFP2 set1) produced an anomalous value for the mTagBFP2 conversion factor, which was 1.67-fold higher than the other replicates. As a result, we present a comparison of the reproducibility of conversion factors as compared with conversion factors from set 2 of the original data (Supplementary Fig. 9C). The full data is provided in Supplementary Table 5. Our data suggest that, generally, the conversion factor values obtained using the described method are highly reproducible (they differ by less than 20% in all cases except for the anomalous mTagBFP2 value, with resultant coefficients of variation between 0.06 and 0.09). Therefore, we recommend users conduct two independent calibrations for each FP, and exercise caution if the replicates differ by over 20%.

### ECmax assay enables calibration without protein purification

Having established that the ECmax assay, a protein quantification assay that relies only on the peak light absorbance of each FP, is highly accurate and sensitive for purified proteins measured using trusted methods, we asked whether this method could enable us to

drop the purification step altogether. Dropping the purification step was not possible for the other assays as they are designed to quantify total protein concentrations, but the ECmax should in principle be specific for the considered FP and may therefore be used to quantify FPs in crude lysates. To investigate this, we harvested and lysed cells expressing our three FPs, separated the soluble fractions and concentrated them. Putting these through an ECmax assay and fluorescence assay, we observed it was possible to quantify FPs in crude lysates with high sensitivity (to 1 ng/μl; Supplementary Fig. 11A), and to obtain almost identical conversion factor values as from our purified FPs, using the mean conversion factor from all (non-anomalous) purifications used as a comparator (Fig. 3e, f, Supplementary Fig. 11B, Supplementary Table 5). Of the two lysis methods tested— sonication and chemical lysis—the former produced more accurate results (conversion factors were closer to the expected conversion factor from purified proteins—90–99% of expected values) and they also had high precision (low variability, with coefficients of variation between 0.02 and 0.12, similar to the CVs observed with purified calibrants). Repeated testing of calibrants prepared by sonication suggests that FPs maintain stability in lysates when stored at 4 °C for a number of weeks (Supplementary Fig. 11C). Using the ECmax assay for FP quantification, it is possible to remove the purification step altogether without compromising calibration accuracy and precision.

### FPCountR results compare favourably to commercial calibrants

We were interested in testing whether our purified FPs gave similar RFU to molecule conversion factors as commercially available FPs. Very few FPs are available commercially, and the majority are green FPs, so we focussed on those. We compiled a table of available GFPs to ascertain the best candidates to order (Supplementary Data File 1). Surprisingly, many were based on first-generation GFPs that preferentially excite in the UV range (unlike GFPmut3 or sfGFP), had incomplete datasheets lacking protein sequence, brightness or spectral information, or were not subject to explicit quality controls (Supplementary Note 3). This suggested their production may be less rigorous than the methods described in this paper and makes them difficult to recommend as calibrants (Supplementary Note 3). Nonetheless, using a commercial TurboGFP, we obtained a relative conversion factor of 95.0% compared to the value obtained using our purified mGFPmut3 (Supplementary Table 5), which suggests that FPs from in-house and (carefully selected) commercial sources may be used interchangeably. Further, calibrations using the small molecule fluorescein produced conversion factors with only 33% error compared to mGFPmut3, provided that the spectral differences between mGFPmut3 and fluorescein were accounted for (Supplementary Fig. 12, Supplementary Table 5), showing decent comparability between a protein and small molecule calibrant. To our knowledge, this is also the first experimental validation that

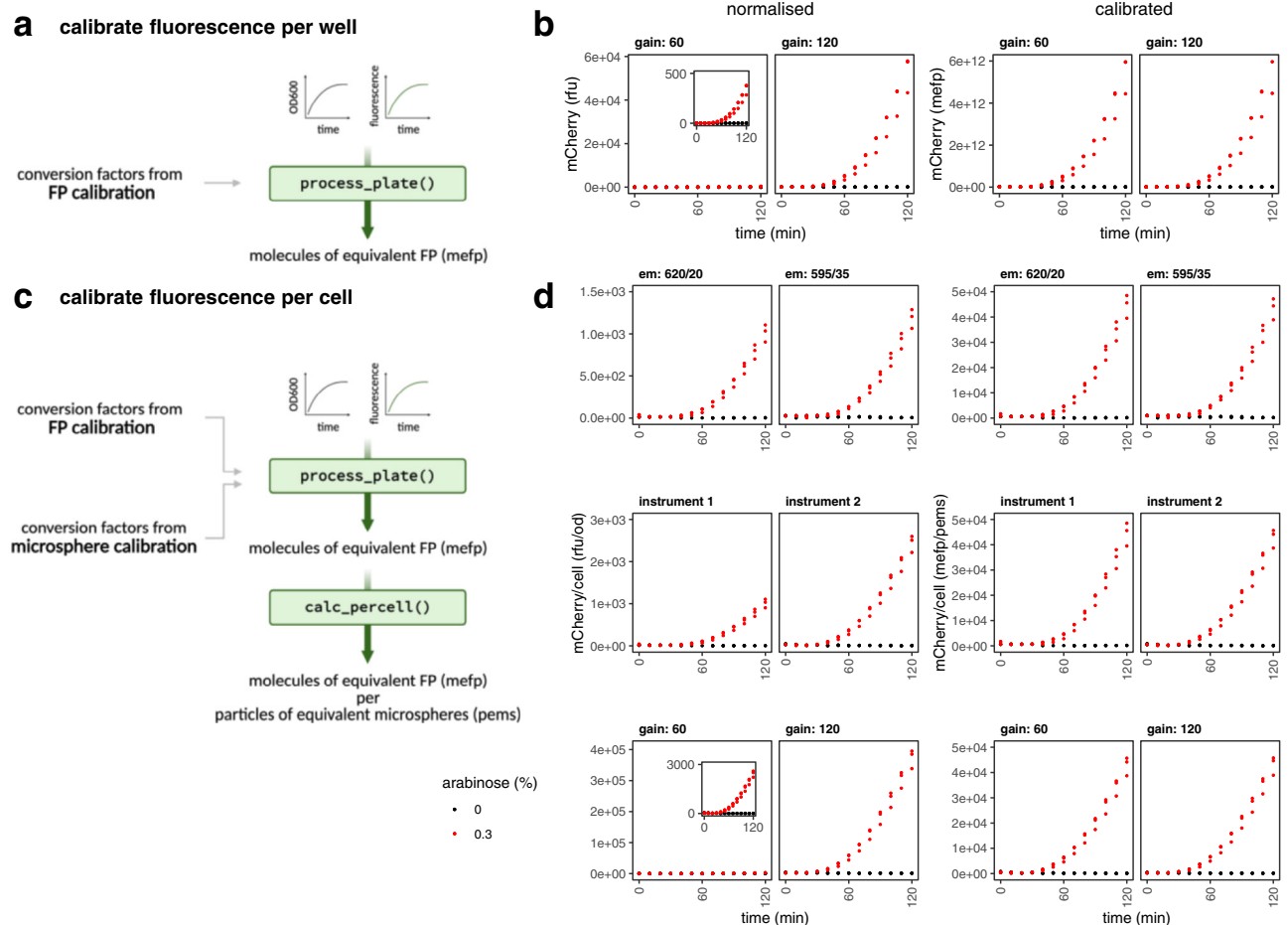

**Fig. 4 | FP calibration allows comparison across gains and instruments for FPs other than GFP. a** Calibration of fluorescence per well in units of MEFP. Time-course experimental data of *E. coli* protein expression may be processed using the process_plate() function using conversion factors obtained from FP calibration. This allows the conversion of normalised data (in relative fluorescence units, rfu) to be converted into calibrated units, of 'molecules of equivalent fluorescent protein' or 'MEFP'. **b** Comparison of normalised vs calibrated data in MEFP. Starter cultures of *E. coli* DH10B containing pS381_ara_mCherry were transferred into a 96-well plate. mCherry expression was uninduced (grey) or induced 0.1% (black) arabinose at 0 min. Absorbance at OD700 and fluorescence was monitored every ten minutes. Data was collected from three biological replicates, each of which is plotted. Left panel: normalised mCherry in units of RFU. Right panel: calibrated mCherry in units

of MEFP. Inset plot shows the same data as the parent plot on a zoomed axis. **c** Calibration of fluorescence per cell in units of MEFP/PEMS. By combining the FP conversion factors with conversion factors from a microsphere calibration the data can be further processed using the calc_fppercell() function into 'per cell' data with units of MEFP per 'particles of equivalent microspheres' (PEMS). **d** Comparison of normalised vs calibrated data in MEFP/PEMS. Expressions were carried out as in (**b**). Normalised and calibrated values are shown when compared across different filter sets (top, notation: emission wavelength/bandwidth), instruments (middle) and gains (bottom). Data for the filter and gain comparisons were taken using the same instrument. Figure panels **a** and **c** created with Biorender.com. Source data are provided as a Source data file.

fluorescein may under certain conditions allow the conversion of RFU not just to molecules of fluorescein but also to approximate molecules of protein.

## Comparison across instruments for FPs other than GFP

While fluorescein enables the comparison of experimental results across laboratories and instruments by converting arbitrary units into 'molecules of equivalent fluorescein' (MEFL) units, FP calibration in principle offers the same capability by converting arbitrary units into units of 'molecules of equivalent FP' (MEFP), which is carried out using FPCountR's process_plate() function (Fig. 4a, b). This allows us to quantify the number of FP molecules in each well of our microplates. To calculate uncalibrated 'per cell' values, typical studies will divide the RFU values by the optical density (OD600 or OD700) of the culture, which quantifies cell density. The calibration of optical density to particle number can be achieved through a similar calibration process using microspheres of similar size to *E. coli*[22,23]. Using both calibrations, it is possible to quantify molecule number per cell in 'molecules of equivalent FP per particles of equivalent microspheres' (MEFP/PEMS,

Fig. 4c), units which should allow cross comparison between different instruments, gains and filter sets. To test this, overnight cultures of *E. coli* containing mCherry expression vectors were split into separate but identical microplates containing arabinose, and were grown in two plate readers using a range of settings. The results show that normalised values of relative fluorescence differ by ~1.5, ~3 and ~130-fold without calibration, whereas such values become reliably comparable after calibration, even for experiments conducted using instrument settings that produce values that cannot be legibly plotted on the same axis (Fig. 4d).

## Estimation of absolute cellular protein concentration

We next asked if calibration to units of MEFP/PEMS was a reasonable approximation for molecule number per cell (Fig. 5a). We carried out microsphere calibration using 1 cm cuvettes and a standard spectro-photometer and obtained conversion factors (Supplementary Table 6) that fell within the range quantified by empirical OD600-specific cell counts[24]. In addition, other authors have confirmed that values of fluorescent protein per cell using fluorescein and microsphere

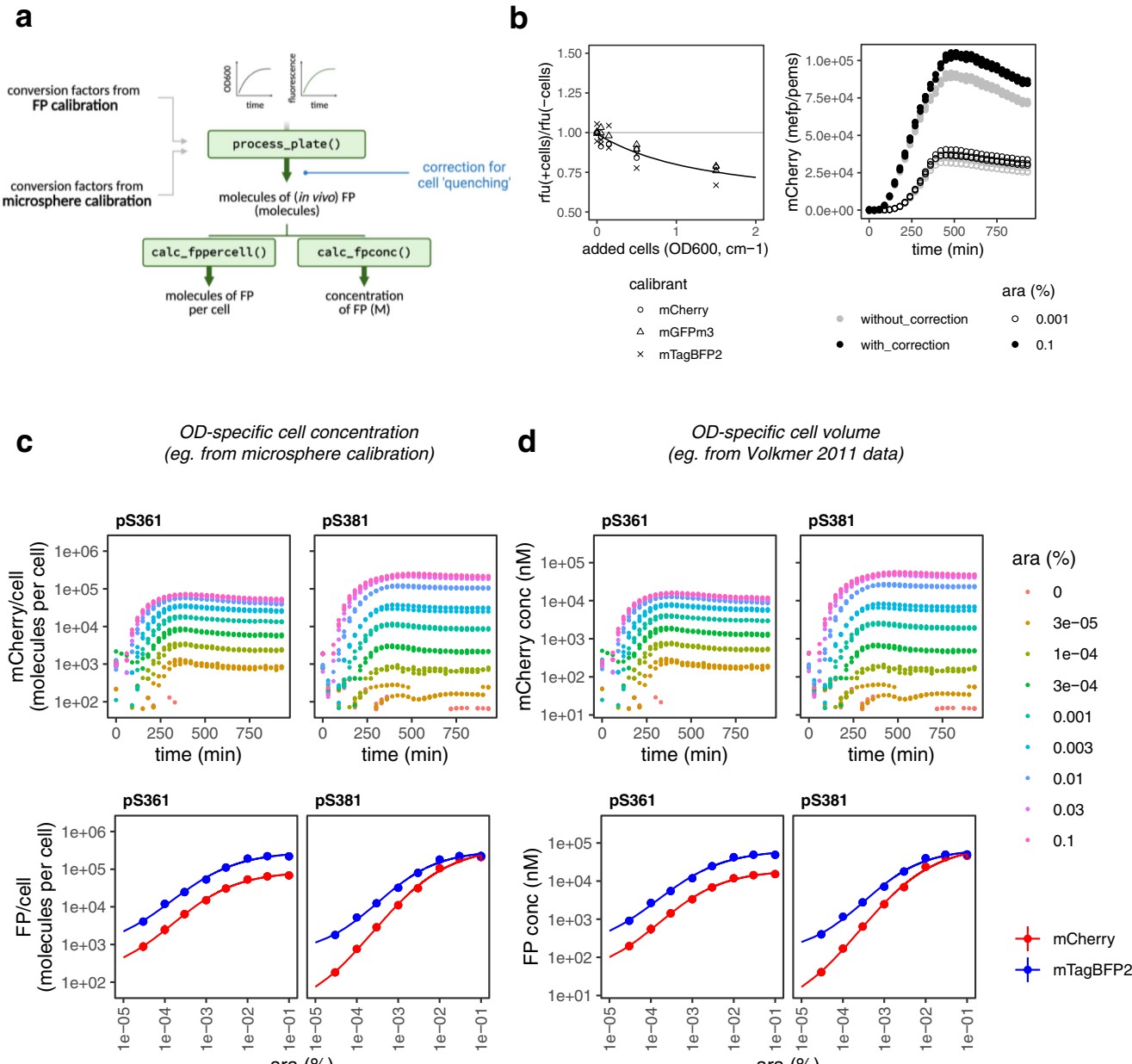

**Fig. 5 | Absolute quantification of *E. coli* timeseries data in molecules per cell.**
**a** Functions to convert experimental data to absolute units. Diagram of modifications to the process_plate() function to (i) incorporate a compensation step based on a quantitative understanding of the impact of cell density on apparent fluorescence (this allows the units to be recorded as molecules per cell), and (ii) to calculate molecular concentration of each FP instead, in molar units.
**b** Quantification of the quenching effect on fluorescence on three FPs. Purified FPs were mixed with non-fluorescent *E. coli* at a range of concentrations, and OD600 and fluorescence intensity were recorded. After normalising for cellular autofluorescence, the fold differences between relative fluorescence intensity (rfu) with (+) and without (−) added cells was quantified (left). Data was collected in duplicates, with both points plotted. A model was fit through this data to enable prediction of expected fluorescence quenching for a given cell density on experimental data. An example of the effect of the correction (right panel). An mCherry expression vector induced with low (open circles) and high (closed

circles) concentration arabinose is presented without (grey) and with (black) correction. Data was collected from three biological replicates and all points are plotted. **c**, **d** Absolute protein quantification in molecules per cell (**c**) and molar concentration (**d**). mCherry expression (top) from medium (pS361, p15A) and high (pS381, colE1) copy vectors was induced at a range of arabinose concentrations and quantified in a timecourse assay in a calibrated plate reader. Data was processed as described in (**a**) and cell estimates based on microsphere calibrations were used to calculate per cell values (**c**), or OD-specific cell volume data from Volkmer et al., 2011 was used to calculate molar concentrations (**d**). Data was collected from three biological replicates, each of which is plotted. (Bottom) mCherry expression from the top panel is compared with mTagBFP2 expression from an identical assay, plotted against arabinose concentration at 420 min post induction. Displayed points show the mean values from two independent experiments, each of which tested three biological replicates. Error bars indicate standard deviations. Figure panel **a** created with Biorender.com. Source data are provided as a Source data file.

calibrants (MEFL/PEMS) are approximately equal to those obtained using fluorescein-calibrated single-cell data on a flow cytometer[7,23], suggesting particle counts were likely to approximate actual cell numbers. However, we observed a major caveat to the use of microspheres as calibrants, which is that their absorbance profiles differ

from that of cells. This can frustrate their ability to provide accurate conversions between 'per cell' data calculated using OD700 versus OD600 measurements (Supplementary Fig. 13 and Supplementary Tables 4–5) for which we discuss solutions in Supplementary Note 4 (and Supplementary Data File 2).

The question of whether the measured fluorescence of FPs in cells is the equivalent of measured fluorescence of the same number of FPs in vitro is less clear. Some authors have found that cells attenuate (or 'quench') fluorescence[15,25], but the magnitude of the effect has not been systematically investigated, particularly for modest cell concentrations found in a typical *E. coli* growth assay. We quantified the quenching properties of *E. coli* cells on our three FPs by mixing an increasing concentration of non-fluorescent cells with purified FPs, and quantifying the difference in apparent fluorescence with added cells (Fig. 5b and Supplementary Fig. 14). Our results suggest that this 'quenching' effect amounts to <20% of the fluorescence signal for moderate cell densities (OD600/cm under 0.5), but increases to about 30% for the highest cell densities typically observed in microplate-scale cultures (OD600/cm around 2). This information was used to add a correction step into the process_plate() calculations so as to compensate for the expected percentage loss of fluorescence with increasing cell density (Fig. 5b, right panel). The complete analytical workflow from calibration to experimental data processing is illustrated in Supplementary Fig. 15.

Using these amendments, it is possible to convert response curve assay data into molecules per cell. Figure 5C shows one experiment using mCherry expression construct in two vectors with different origins of replication. Using these vectors, we obtain figures for mCherry abundance that vary between about 900 to 70,000 molecules per cell for p15A, and 200 to 200,000 for colE1. Protein abundance information, available from proteomics and ribosome profiling studies, suggests that the typical *E. coli* protein is present in the order of $10^2$–$10^3$ copies per cell, and the most abundant are present in the order of $10^5$ copies per cell or higher[26–28]. Over-induction using the high-copy (colE1) vector therefore appears to allow synthetic protein expression to reach the level of the most abundant proteins in the cell. This is supported by the fact that these vectors reliably overexpress FPs to a level observable by SDS-PAGE in unpurified lysates (Fig. 1, Supplementary Fig. 1). Modest expression ($10^2$–$10^3$ per cell) can be achieved by combining low arabinose concentrations with either vector. In other words, the colE1 vector allows us to utilise the full spectrum of protein abundances from modestly expressed enzymes such as the RecBCD helicase (~100 copies per cell) through to the most abundant ribosomal proteins (~100,000 copies per cell, ref. 28). We could also use this to compare the number of molecules produced from the same vectors but two different FPs. Interestingly, measurements from identical SEVA vectors revealed that while the FP abundances were in the same order of magnitude, mTagBFP2 accumulated to higher levels per cell than mCherry by 3.8-fold on average, despite sharing the same promoter, 5' untranslated region, ribosome binding site and N-terminal protein sequences (Fig. 5c, d). This could be due to translation rate effects from RNA level effects in the coding sequences beyond the first 11 standardised codons, or else due to differences in degradation kinetics between the two proteins.

In 2011, Volkmer and colleagues noted that while OD-specific *E. coli* cell counts varied with growth rate, the OD-specific total cell volume was ~3.6 μl per OD600/cm, regardless of strain or growth condition[24]. Using OD as a measure of the cumulative cellular volume in a culture could therefore be used to convert fluorescence and OD measurements into concentrations in molar units, instead of 'per cell' values, and such conversions may be more appropriate for comparing experimental results with quantitative modelling of cellular reaction networks, since they are unaffected by growth rate differences. Using this method, we found that FP abundances using the same vectors populate a range of concentrations between 0.01 and 100 μM (Fig. 5d).

### Revealing the hidden properties of fluorescent proteins

Finally, we were interested in testing whether absolute quantification could illuminate a well-known source of error in bacterial assays. The presence of red FPs has been suggested to interfere with bacterial cell density estimations at 600 nm since red FPs typically absorb well at this wavelength[29] and has led to the conclusion that circuits using red FPs must be quantified at 700 nm, which is unaffected by their presence. However, the number of molecules of red FPs that might be required for this effect to occur has never been quantified.

Calibrated timecourse data of mCherry overexpression in *E. coli* (Fig. 6) was examined to quantify these effects, with mGFPmut3 and mTagBFP2 used as negative controls. The ratio between OD600 and OD700 measurements was used to identify errors caused by red FP absorbance. Linear models fitted to the relationship between measured OD600 and OD700 values confirmed that this relationship was very similar for all uninduced cells (Fig. 6b; OD600 = 1.30 * OD700 − 0.02), but mCherry induction resulted in a measurable deviation (OD600 = 1.37 * OD700 − 0.03, Fig. 6b, c). Looking at the relationship between this shift and cellular protein copy number (Fig. 6d), our results indicated that the OD600 error for mCherry was only apparent when mCherry levels per cell were high (over 100,000 per cell), and that the magnitude of this error was only about 5%. In contrast, mGFPmut3 expression had no effect on cell density estimation using OD600, as expected. Surprisingly, we observed the opposite trend for mTagBFP2, in which OD600 measurement appeared to underestimate cell density where mTagBFP2 was expressed at high levels per cell. The reasons for this are currently unclear and beyond the scope of this paper, but our results suggest that this could be an interesting avenue for future work. Generally, this experiment confirmed that the presence of low to moderate FP levels per cell (under 100,000) do not perturb cell density estimates, and errors are of a lower magnitude in all cases than those from cellular fluorescence quenching.

## Discussion

Our aim for this work was to develop a generalisable method that allows fluorescence readings on microplate readers to be calibrated to molecular units of fluorescent protein. The method ought to be (1) accurate and sensitive, (2) as simple as possible, (3) suitable for any fluorescent protein, and (4) easily analysed. To develop the method, we adopted the principles of redundant experimental design, including the validation of multiple assay types, characterisation of the method's consistency, and the assessment of its generality for three different FPs[30].

Our initial method using a simple purification protocol (Fig. 2) and a commercial protein assay allowed us to develop an analysis pipeline to obtain conversion factors from purified FP calibrants. To demonstrate the accuracy and validity of our calibrations, we verified that the absorbance and fluorescence spectra of the calibrants matched their counterparts on FPbase (Fig. 1, Supplementary Fig. 2; refs. 21, 31), and validated our initial protein assay measurements by cross comparison with two further methods. To do this, we adapted the low-throughput A280 assay into an accurate, high-throughput assay format, and showed that these were suitable for use with FPs even though some absorb in the near-UV range. This type of assay for FP quantification has not, to our knowledge, been demonstrated in the literature before, and we contend that it is likely to be of particular interest since it requires no calibrant or commercial reagent, no expensive quartz-based consumables and exhibits a sensitivity that exceeds that of commercial systems such as the Nanodrop.

We also discovered a methodological shortcut to obtaining FP concentrations using the extinction coefficients at their maximum excitation wavelength, the ECmax assay, which was both the simplest and the most robust of all the assays tested. Specifically, the ECmax assay was the least affected by buffer conditions, and had the largest linear range (to almost 0.1 ng/μl; Fig. 3). We note that the assay is limited by the fact that it requires the used FP to be documented on FPbase, and assumes that the documented ECmax measured by other laboratories is accurate. Promisingly, our results suggest good inter-lab agreement for these measurements (compare A280 and ECmax

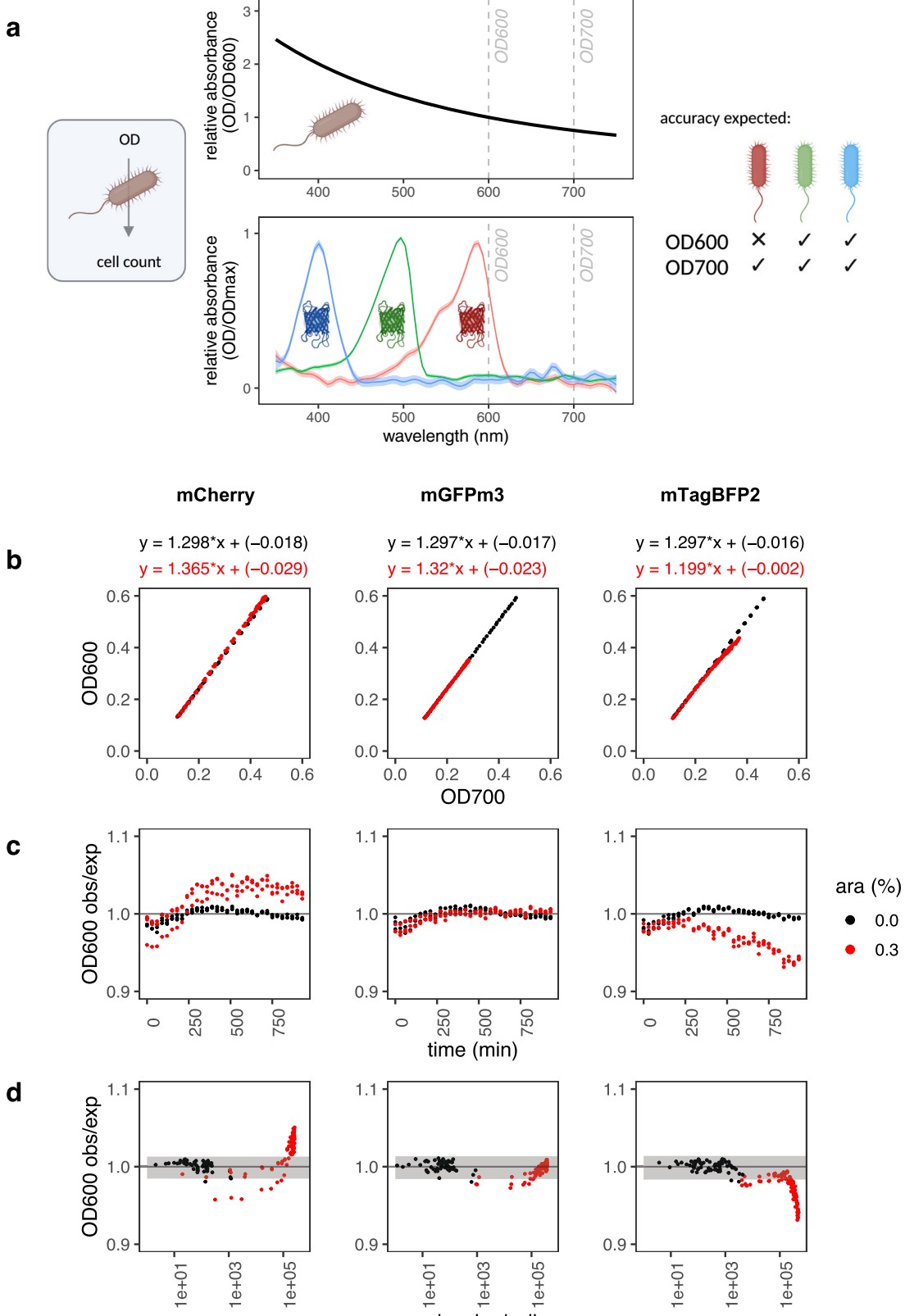

estimates, Fig. 3c). In addition, an analysis of all FPs on FPbase, comparing the proteins' extinction coefficients at 280 nm versus at their maximum excitation wavelength, supports the idea that the ECmax is a more sensitive assay for most FPs (Supplementary Fig. 16, Supplementary Table 4). While the EC(280) values are highly uniform (median: 27,400 M$^{-1}$ cm$^{-1}$), likely because most FPs are very similar in

size, the ECmax values are mostly considerably larger (median: 64,200 M$^{-1}$ cm$^{-1}$). Impressively, as the ECmax assay specifically quantifies FP concentration rather than total protein concentration, we were also able to show that it is possible to do these calibrations in crude lysate without compromising on accuracy, demonstrating that calibrants may be produced without affinity purification (Fig. 3d–f).

**Fig. 6 | Evaluation of OD600:OD700 ratios in cell growth assays. a** Cell count accuracy with FPs. In bacterial assays, cell counts are obtained from calibrated OD600 or OD700 measurements on the assumption that the only contributor to absorbance at 600 or 700 nm are cells. Cellular OD600:OD700 ratios (top plot, data from Supplementary Fig. 11) approximate 1.3. Red FPs like mCherry absorb light at wavelengths used for cell density assessments (bottom plot, data from Supplementary Fig. 1B) and may lead to an error in cell count estimates. **b–d** Timecourse expression assays of three FPs from the pS381 vector were monitored for OD600, OD700 and fluorescence intensity every 30 min, without (black) and with (red) arabinose. Data was collected from three biological replicates, and all points are plotted. This is a representative experiment of at least two independent

experiments for each FP. Note OD600:OD700 ratios once again approximate 1.3. **b** Relationship between OD700 and OD600 values. Linear models were fitted to data from three FPs, and shown above each plot. **c** OD600 error plotted against time. OD600 error was obtained by dividing observed values by expected values. Expected values were calculated from the OD700 values and the measured OD600-OD700 relationship of the 0% arabinose sample (see **b**). **d** Effect of FP abundance on OD600 error. The OD600 error was plotted against abundance of each FP in molecules/cell. Grey shading indicates the mean of the samples without arabinose ± 2*standard deviations. Figure panel A created with Biorender.com. Source data are provided as a Source data file.

---

Overall, we expect calibrations carried out using the ECmax method, using either purified calibrants or (sonicated) cell lysates, to be equally accurate. We provide both rigorous and expedient protocols for prospective users on protocols.io.

While fluorescent proteins are biological molecules whose fluorescence activity is dependent on their condition of production and testing (such as pH and the availability of oxygen), the use of a few well-established techniques from standard bacterial protein overexpression protocols allows such problems to be avoided. For instance, the expression protocol we describe is designed to produce high levels of protein: we use high copy vectors and long expression times. However, we grow cells at low temperatures of 25–30 °C for the expression, which minimises the chance that overexpressed proteins will misfold or aggregate[32]. The presence of aggregates can be checked by SDS-PAGE after the clarification step that separates the soluble and insoluble fraction[32]. We rarely see significant aggregation (Fig. 2c). Oxygen is also required for chromophore maturation[19], however shaken flask cultures will be well aerated, and the lysis and/or purification procedure gives time for any remaining immature proteins to mature during the protocol itself. (Simply exposing cells to air allows maturation of FPs expressed in anoxic conditions[33].) Furthermore, the buffer used for the calibration assays should match the cellular pH (we use pH 7.5). We also conduct calibration assays at a temperature that matches our bacterial assays (though we do not expect that minor temperature changes would have a significant effect on FP brightness or behaviour).

Current users of fluorescein calibrations may be interested in how this method compares with calibration using a serial dilution of fluorescein. The protocol for fluorescein calibration is certainly cheaper (£0.23 per calibration using details from ref. [6]) owing to the fact that fluorescein is a low-cost fluorophore, and simpler, as the use of commercial calibrants obviates the need for calibrant preparation or concentration determination. The key advantage of FP calibration lies in its ability to be used to directly convert arbitrary units directly to molecules of one's desired FP, coupled with the freedom to produce bespoke calibrants from any desired FP, or indeed multiple FPs, from across the spectrum, whether or not they match the properties of fluorescein. While commercial FPs cost an average of £39.05 per calibration (assuming use of 10 μg FP per calibration, Supplementary Data File 1), the estimated cost of producing one batch of FP calibrants in lysate is only £10.18, from which the standard yields are approximately 100 μg FP (using only a fraction of the culture). While FPs can display condition-dependent fluorescence, as discussed above, these effects can be minimised: indeed, they can be used to test the temperature or pH sensitivity of a user's FP under controlled conditions. Furthermore, fluorescein and other small molecules also display condition-specific fluorescence, though this point is rarely considered[34]. While it is difficult to directly compare the precision of this method with published data on fluorescein, the FPCountR protocol is clearly highly reproducible. Calculated conversion factors are typically within ±10% of the expected value and coefficients of variation between independent replicates are typically below 0.1 (Supplementary Table 5). Following on from the exemplary work on fluorescein[6], there is clear potential

for future inter-lab studies to extend our understanding about the accessibility and reproducibility of the FPCountR protocol.

Such calibrations can then be used, first, to enable the comparison of experimental results from different plate readers or across different settings, in molecules of fluorescent protein per particles of equivalent microspheres (MEFP/PEMS; Fig. 4b). This is akin to using fluorescein but with a broader application range, since using bespoke calibrants for each FP allows us to calibrate instruments for any FP regardless of its spectral characteristics. If used merely as comparative units, the precision (repeatability) of each calibration is important, but their accuracy (whether conversions predict molecule numbers as closely as possible) is not. Second, they can be used to express protein abundance as 'molecules per cell' (Fig. 5c). Accuracy here is an important consideration, and will not only depend on the accuracy of the FP calibration (discussed above), but also on the microsphere calibration, and the removal of any interactions between absorbance and fluorescence characteristics of cells expressing FPs. As to microsphere calibration, cross-comparison with flow cytometry data suggests cell count estimates from microsphere calibrations are reasonable[7,23], although Beal and colleagues used 0.961 μm microspheres whose size is closer to *E. coli* than those used in this study (0.890 μm; the larger type are now unavailable). Our protein abundance estimates ($10^2$–$10^5$ proteins per cell; Fig. 5b) are also within reasonable bounds[26,28], corroborating their use, and indicating that FP calibrations may enable protein abundance comparisons between microplate assays and proteomics experiments.

The consideration of whether the presence of cells interferes with fluorescence quantification (or vice versa) is multifaceted. It is well-known that cells interfere with fluorescence measurements through autofluorescence. Cellular autofluorescence is known to largely impact GFP quantification accuracy[35], and is corrected for in FPCountR by normalising to the background fluorescence of control cells at a similar OD (akin to refs. [7, 15]). The 'quenching' of apparent FP fluorescence by the presence of cells is more rarely considered[14,15]. We have found that the effect size is comparable for different FPs (Supplementary Fig. 14), unlike for autofluorescence, supporting previous observations by others[25]. A correction for this attenuation in FPCountR compensates fluorescence according to the expected percentage quenched at the measured OD (Fig. 5b). Both of these corrections are included in the process_plate() function.

Certain sources of error cannot be adequately addressed by calibration alone. While some have noted that pH can affect the molecular brightness of certain FPs[15], this could not be compensated for analytically without user input detailing both the pH response profile of the included FPs and the pH of their cells. Fortunately, since the cellular pH in *E. coli* is limited between pH 7.2–7.8[36], and even pH-sensitive FPs exhibit only mild (<10%) variation in molecular brightness for pH values between 7 and 8[37,38], pH-dependent changes in molecular brightness are unlikely to have a large effect on quantifications.

The overall error in the accuracy of protein per cell quantifications using FP- and microsphere-calibrated microplate readers can be estimated as the sum of its components: 20% (from protein quantification error, Fig. 3c, Supplementary Tables 1 and 2) and 33% (from cell count

error deduced from flow cytometer counts, Fedorec et al., 2020), which results in approximately 50% error. This should be accurate enough to allow protein per cell quantification not just to the correct order of magnitude (ie. to estimate whether protein abundance is in the hundreds per cell or thousands per cell), but also suggests that quantifications are likely to be accurate to within two-fold of the true values. This should be more than adequate for most applications, including estimations of the relative magnitude of two or more FPs, as well as for genetic circuit modelling. Additional errors, such as those from cell quenching or from the comparison of OD600- vs OD700- quantified cell counts, can be avoided by the use of the FPCountR analytical R package. Others, such as those from pH-sensitive FPs, can be avoided by using pH-insensitive FPs.

Overall, it seems likely that the accuracy of calibrated molecule counts per cell may be limited by the accuracy of cell count calibration, since it is known that OD-specific cell counts change with growth rate, and that this is due to the positive relationship between growth rate and cell size for *E. coli*[24]. We therefore expect these calculations of protein counts per cell will be more accurate where cell size and microsphere size are as similar as possible, and inaccuracies will arise if the differences become significant. Thus, these values are likely to be approximately accurate so long as the following assumptions are true: that *E. coli* absorbance (scatter) is well represented by microsphere properties, and that it doesn't change significantly over time or between samples. These will likely to be true for many *E. coli* experiments since cell shrinkage has been shown to take place only after several hours in stationary culture, and only large growth rate differences maintained for prolonged periods were observed to make a significant difference to cell size in exponentially growing cells[24], but may fail if circuits impose significant burden on host cell resources that impacts growth rate or cell size[39–41]. In contrast, using a conversion from OD to total cell volume in a given culture well allows us to remove the requirement for counting cells (Fig. 5d) and allows the expression of protein concentrations as 0.01–100 μM. These figures should be approximately accurate under the assumption that the OD-specific cell volume doesn't vary significantly between samples or over time. This is strongly supported by the results of Volkmer and colleagues[24] whose data suggests this variation is within 2-fold across a wide range of growth conditions, but also by others who have shown that as cell volumes increase, their OD-specific cell counts decrease by approximately the same magnitude[39,42]. Units of concentration may also be more meaningful for reaction modelling since ultimately it is molecular concentration that is critical for binding and kinetics[9,43,44].

We used absolute protein quantification to investigate the problem first described by Hecht and colleagues[29] in which an association between mCherry overexpression and deviations in cell density measurements were made. Repeating these assays showed a clear effect, but this effect was of modest magnitude (<5%) and was only apparent at very high mCherry levels per cell (over $10^5$ molecules per cell; Fig. 6). This suggests that for most circuits that use moderate expression levels to minimise cellular burden, OD600 values would remain an accurate way to quantify cell density. We also unexpectedly observed the opposite effect in mTagBFP2 expressions. As this analysis was technically a quantification of whether OD600 and OD700 measurements deviate from each other in the presence of different FPs, these results might not suggest that OD600-based cell density readings in the presence of high mTagBFP2 are inaccurate. It might instead suggest that, conversely for mTagBFP2, the OD700 readings are inaccurate. We do not currently have an explanation for this finding. One possible contributing factor may be that while it is often described as one of the best blue FPs, mTagBFP2 has a higher propensity to aggregate than mCherry and most GFPs[45]. Cell stress is known to induce *E. coli* to elongate[39,46], which may affect its scattering properties. It is possible that very high FP levels may frequently cause a small but significant error in cell density estimates due to combinations of effects from light absorption and scatter that warrants more study in order to allow us to further improve molecular quantifications of FPs under those conditions.

While flow cytometry and mass spectrometry allow us to probe single-cell measurements or the protein complement of an entire cell, respectively, microplate readers remain an important screening platform in the Design-Build-Test-Learn cycle due to ease of use, low cost and high iterative capabilities. This necessitates the development of methods for extracting informative numbers from such data. The ability to extract absolute protein abundance information from assays of engineered microbial cells is imperative to enable the characterisation, optimisation and tuning of genetic circuits in a rigorous and quantitative manner, and will allow for a deeper insight into how protein abundance affects genetic construct behaviour, cellular burden and growth rate. Importantly, our approach enables accurate, non-destructive, and easy protein abundance comparisons, even between samples that differ in growth rate or cell size.

Further, such absolute quantification need not be limited to fluorescent proteins. The last few years have seen a fantastic expansion of fluorogenic molecules, tools that have allowed the specific quantification of localised proteins[47–49], proteins in anaerobic environments[50,51], and the fluorescent quantification of RNAs[44,52,53]. Calibration of these molecules would be more complex to achieve but no less valuable. Equally, calibration of alternative instruments such as flow cytometers are also of interest, but would require a very different approach, requiring calibrants to be attached to particles of cell-like sizes. We hope that our demonstration of how to achieve absolute FP quantifications from microplate reader assays using FPCountR can contribute to the effort to develop more quantitative approaches for the analysis of circuit behaviour in synthetic biology and beyond.

## Methods

### Materials
Primers and gblocks were obtained from IDT, and *E. coli* strain DH5-alpha (Invitrogen, 18265-017) was used for molecular cloning. Chemicals and protein reagents were purchased from Merck, Sigma, ThermoFisher Scientific, and Bio-Rad, molecular biology reagents from NEB and Life Technologies and general laboratory reagents from Corning, Greiner Bio-One, Starlab and VWR.

### Fluorescent proteins
The mCherry[54] protein sequence was based on the FPbase entry for mcherry (https://www.fpbase.org/protein/mcherry/) with the following changes: M1(MVHHHHHHGSG). The mGFPmut3[55] protein sequence was based on the FPbase entry for gfpmut3 (https://www.fpbase.org/protein/gfpmut3/) with the following changes: M1(MVHHHHHHGSG), A206K (a substitution to make the protein monomeric; refs. 18, 19). The mTagBFP2[56] protein sequence was based on the FPbase entry for mtagbfp2 (https://www.fpbase.org/protein/mtagbfp2/) with the following changes: M1(MVHHHHHHGSG). Full protein sequences are provided in Supplementary Data File 3.

### DNA assembly
Vectors for fluorescent protein purification and growth curve assays were constructed according to standard protocols, via Golden Gate and Gibson assembly techniques using *E. coli* strain DH5α (Invitrogen, 18265-017). Constructs were assembled into Standardised European Vector Architecture (SEVA) backbones[57]: pS381 was generated from pS181 with chloramphenicol substitution; pS361 was generated similarly from pS161. In an effort to approximately equalise expression levels between different proteins, the 5′ untranslated region (including ribosome binding site) and 5′ region of each construct was set to be identical up to residue 11 (coding region begins: *DNA*: ATGGTTCACCA TCATCATCACcacGGTtcgggc, *protein*: MVHHHHHHGSG). The second

residue was set to valine to reduce the effects of N-end mediated degradation[58]. The affinity tag chosen for purification was the His$_6$ tag, which was followed by a short unstructured linker (GSG). Full DNA sequences of vectors are provided in Supplementary Data File 3. All three pS381 plasmids used in purifications are available from Addgene under the IDs 186733–186735.

## FPCount (wet lab) protocol

The FPCount protocol consists of calibrant preparation followed by a protein assay (for protein concentration) and a fluorescence assay (for protein activity). Each step is described in the following sections below. In addition, we have provided step-by-step instructions for the recommended FPCount protocol in Supplementary Note 5. This protocol consists of the preparation of calibrants as fluorescent proteins in cell lysates prepared by sonication, and the use of the ECmax assay as the protein assay. This protocol is also available on protocols.io, at https://www.protocols.io/view/fpcount-protocol-in-lysate-purification-free-proto-bzudp6s6[59]. We also detail two other calibration protocols on protocols.io: the full protocol conducted for Fig. 3, using FP purification and three protein assays for cross-validation, available at https://www.protocols.io/view/fpcount-protocol-full-protocol-bztsp6ne[60], and a shorter protocol requiring purification but using the ECmax assay only, available at https://www.protocols.io/view/fpcount-protocol-short-protocol-bzt6p6re[61].

### Protein expression and harvesting

Fluorescent proteins were produced using pS381 (SEVA) expression vectors in *E. coli* BL21(DE3) strains. Glycerol stocks were inoculated into 50 ml Luria Broth (Miller) supplemented with 50 µg/ml chloramphenicol and 0.02% arabinose and were grown overnight at 30 °C at 250 rpm. Cells were harvested after ~16 h by transferring them to pre-chilled containers on ice. All further steps were conducted on ice. OD600 readings were taken, and 40 OD of cells was transferred to fresh falcon tubes, washed once in T50N300 (50 mM Tris-HCl pH 7.5, 300 mM NaCl) and resuspended in lysis buffer (T50N300, 1X protease inhibitors (EDTA-free, Pierce A32955), filter sterilised, supplemented with lysozyme 100 µg/ml). Cells were separated into 20 OD (2 ml) fractions and sonicated (QSonica Q125 sonicator, 50% amplitude, 10 s on, 10 s off, 2 min). Lysates were supplemented with 5 mM CaCl$_2$, 50 mM MgCl$_2$ and treated with DNase I (50 U/ml, bovine pancreas, MP Biomedicals, 219006210) for 30 min at 4 °C. Soluble fractions were isolated by isolation of the supernatant after centrifugation (16,000 × *g*, 30 min, 4 °C), and both fractions were checked by SDS-PAGE followed by staining with Coomassie-based dye according to the manufacturer's instructions (Instant Blue Protein Stain, Sigma ISB1L-1L).

### Protein purification

Fluorescent proteins were purified in batch using His-tag affinity chromatography at room temperature according to the resin manufacturer's instructions (Thermo Fisher). Lysates were supplemented with 10 mM imidazole and 600 µl was applied to HisPur Cobalt resin (300 µl, ThermoFisher) equilibrated in Binding Buffer (T50N300 + pi, 10 mM imidazole), mixed and incubated at room temperature for ~15 min before removal (1000 × *g*, 1 min). This was repeated four times, before the resin was washed with 10 column volumes of Binding Buffer. Protein was eluted in Elution Buffer (T50N300 + pi, 150 mM imidazole). All protein fractions for calibration were stored protected from light at 4 °C.

### Preparation for calibration assays

Elution fractions were combined and concentrated approximately 10-fold using Amicon centrifugal filter columns (Merck, UFC5010), followed by buffer exchange (1000x) into T5N15 (5 mM Tris-HCl pH 7.5, 15 mM NaCl) or T5N15 + pi (T5N15, 1x protease inhibitors, filter sterilised).

### Microplate reader assays

All assays were carried out using a Tecan Spark microplate reader (using SparkControl Magellan V 3.1 software) except the fluorescence spectra assays, which were carried out using a BMG Clariostar Plus microplate reader (using CLARIOstar v5.60 and MARS v3.40 software).

### Calibration assays

For each FP calibration, both concentration and fluorescence assays were carried out on the same dilution series of protein. Concentrated, buffer exchanged FP (100 µl) was diluted in 900 µl buffer, then diluted 1:2 into 500 µl buffer in 1.5 ml eppendorfs. A total of 11 dilutions were prepared in this way, distributed into UV-transparent microplates (Greiner, 655801) as duplicates (225 µl). Bovine serum albumin (BSA) standards (from Micro BCA Protein Assay kit, ThermoFisher, 23235) were prepared in parallel with the same buffer(s). This dilution set was then subjected to the protein concentration and fluorescence assays.

### Protein concentration assays: A280 assay and ECmax assay

Absorbance assays were carried out on 225 µl protein in UV-transparent plates, using the Spark absorbance scan method (see Supplementary Methods).

### Fluorescence assays

Following absorbance scans, 200 µl from each well of the original plate was transferred into clear polystyrene plates (Corning, 3370). This plate was sealed (Eppendorf Masterclear real-time PCR film adhesive, 30132947) and used to run the Spark fluorescence methods (see Supplementary Methods) on all relevant instruments, channels and gains.

### Protein concentration assays: microBCA assay

BCA assays were carried out using the Micro BCA Protein Assay kit (ThermoFisher, 23235) according to the manufacturer's instructions (microplate protocol). Briefly, 150 µl of working reagent was dispensed into a clean microplate, and 150 µl from each well of the fluorescence assay plate was mixed into the reagent with a multichannel pipettor. Reactions were covered with a plate seal (BreatheEasy sealing membrane, Sigma, Z380059), and subjected to the Spark microBCA method (see Supplementary Methods).

### Fluorescein calibration

Fluorescein (Thermo Fisher R14782, 1 mM in DMSO) was diluted to a 100 µM stock solution in 100 mM NaOH (the appropriate buffer for fluorescein). This stock (100 µl) was diluted in 900 µl buffer, then diluted 1:2 into 500 µl buffer in 1.5 ml eppendorfs. A total of 11 dilutions were prepared in this way, distributed into clear polystyrene plates (Corning, 3370) as duplicates (200 µl). This plate was sealed (Eppendorf Masterclear real-time PCR film adhesive, 30132947) and used to run the Spark fluorescence methods (see Supplementary Methods) on the appropriate filter set (ex: 485/20, em: 535/35) on a Tecan Spark plate reader. Data were processed using FPCountR-type functions with or without normalisation for relative brightness.

### Relative brightness normalisation

To compare conversion factors from different FPs or small molecules, we calculated the relative brightness of each molecule to be compared. This calculation attempts to normalise for the fluorescence characteristics that differ between calibrants, namely (i) their brightness, and (ii) how well their excitation and emission spectra overlap with the relevant instrument filter sets. The efficiency of excitation in the excitation filter (480 nm with 20 nm bandwidth) in exciting the fluorophore was taken as $ex.eff_{480/20} = \sum_{i=470}^{490} A_{\lambda=i}$ where absorbance ($A$) values for that at every wavelength ($\lambda$) in the excitation filter's bandwidth (in 1 nm steps) were summed, using normalised absorbance

spectra where the maximal absorbance is set to 1. The efficiency of the emission filter (325 nm with 35 nm bandwidth) in detecting fluorescence from the fluorophore was defined as: $em.eff_{535/25} = \frac{\sum_{i=522.5}^{547.5} em_{\lambda=i}}{\sum em_{\lambda=i}}$, where the sum of the emitted fluorescence at the relevant wavelengths was taken and normalised to the sum of the total fluorescence over the entire emission spectrum. Again, spectra used were normalised spectra where the maximal values had been set to 1. Finally, relative brightness was taken as: $relative\ brightness = EC * ex.eff_{480/20} * QY * em.eff_{535/25}$ where $EC$ is the extinction coefficient ($M^{-1}\,cm^{-1}$) and $QY$ is the quantum yield. Conversion factors ($CF$, in RFU/molecule) were converted to normalised conversion factors ($CFnorm$) as: $CFnorm = CF * \frac{relative\ brightness_{mGFPmut3}}{relative\ brightness_{calibrant}}$.

### Calibration of OD600 and OD700 values using microspheres

Calibration of optical density readings used to quantify cell number (OD600 and OD700) was carried out according to published protocols[7]. The microspheres used were monodisperse silica microspheres (Cospheric, SiO2MS-2.0, 2.0 g/cc, d50 = 0.890 μm, CV = 3.2%, <1% Doubles).

### Bacterial timecourse assays

DH10B *E. coli* transformants were grown overnight in M9 medium (M9 salts (1X, Sigma M6030), casamino acids (0.2%), fructose (0.8%), thiamine HCl (0.25 mg/ml), $MgSO_4$ (2 mM), $CaCl_2$ (0.1 mM)) supplemented with 50 μg/ml chloramphenicol, in a deep-well plate (30 °C, 700 rpm), and diluted the following morning into fresh M9 with antibiotic (deep-well plate, 30 °C, 700 rpm) to an OD600 ($cm^{-1}$) of 0.05. After 1 h, cultures were transferred into clear 96-well microplates (Corning, 3370) with pre-loaded arabinose (5 μl). Plates were sealed (BreatheEasy sealing membrane, Sigma, Z380059) and grown in a Tecan Spark plate reader in kinetic mode (see Supplementary Methods, Spark growth curve method).

### Statistics and reproducibility

All SDS-PAGE results presented in this manuscript are representative of a typical experiment of that kind, with at least two but usually more independent repeats showing similar results.

### Analytical methods

All data were analysed using R version 4.0.3[62]. The FPCountR package that was developed for the FP calibrations is available on GitHub at https://github.com/ec363/fpcountr[63]. Supplementary Note 6 includes a description of the analytical steps of the key functions. For a summary of the functions, see Fig. 1 and Supplementary Fig. 15.

### Fluorescence scans

Excitation and emission spectra of the fluorescent proteins were conducted using a BMG Clariostar Plus microplate reader in sealed plates (Corning, 3370; Eppendorf Masterclear real-time PCR film adhesive, 30132947) at wavelengths appropriate to each FP (see Supplementary Methods).

### Figures

Figures were created using RStudio and Biorender.com.

### Reporting summary

Further information on research design is available in the Nature Research Reporting Summary linked to this article.

## Data availability

This study includes no data deposited in external repositories. Key datasets are provided in the Supplementary Tables. Further datasets are provided in the Source data file provided with this paper. External data from FPbase was taken from https://www.fpbase.org/. Plasmids generated for FP purification are available on Addgene. Other plasmids generated in this paper are available without restrictions from the authors on request. Source data are provided with this paper.

## Code availability

Computer code produced in this study is available on GitHub (https://github.com/ec363/fpcountr).

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

## Acknowledgements
G.B.S. and E.C. acknowledge funding from the Royal Academy of Engineering (RAEng CiET 1819\5).

## Author contributions
E.C. and G.B.S. conceived the study. E.C. conducted the experiments, analysed the data and wrote the R package. E.C. and G.B.S. wrote the manuscript.

## Competing interests
The authors declare no competing interests.
