## [Peer Review File · Nature Communications]

Absolute protein quantification using fluorescence measurements with FPCountRReviewer #1 (Remarks to the Author):

The authors of this manuscript propose an alternative to the iGEM plate reader calibration assay. In their protocol, the authors exchange small-molecule fluorescent dyes for fluorescent proteins as a calibrant. Fluorescent dyes with a well-matched spectrum already provide a usefully accurate estimate of fluorescent protein molecule count. By switching the calibrant from a proxy molecule to the actual molecule of interest, however, the authors' assay has the potential to allow an increase of the accuracy of estimates, which will in turn increase the comparability and reusability of data. The authors approach, like the iGEM approach, is clearly superior to normalization against a "standard" strain, in this case because the lysing of cells and use of absorption to quantify protein count mitigates most aspects of cell culture variation (but not all, per below).

The authors provide strong evidence that their protocol is indeed able to provide reasonably accurate estimates of fluorescent protein count, as well as making use of this approach to evaluate the magnitude of impact that cell density and protein absorbance have on protein count estimates. It is not yet clear, however, whether the iGEM approach or the authors' approach has better efficacy.

Overall, this is interesting work and deserving of publication once a few key issues are addressed, these being:

1) Lack of a head-to-head comparison between the iGEM methods the authors cite and their own methods. While the authors are aware of these assays and claim an increase in accuracy over the iGEM methods, they do not actually demonstrate this. In particular, a comparison with flow cytometry data would be valuable, in order to determine the actual accuracy of the fluorescence/cell estimates produced by the authors. Calibration beads with known quantities of GFP and mCherry are both commercially available, and can be used in the same manner as the MEFL-calibrated beads in the iGEM studies.

2) Quantification of precision: the authors make many qualitative statements about the efficacy of their assay, but never provide any quantification of the precision of their approach across independently prepared calibration curves. While an interlaboratory study like the iGEM study would be ideal, that is clearly out of scope for the current manuscript (though it would be good to see it discussed as potential future work). The authors must, however, provide numerical quantification of the precision of their results across multiple independently prepared replicates, in order to provide a direct numerical comparison of the precision of the authors' method and the iGEM method.

3) Potential impact from using a biological calibrant as opposed to an independent calibrant. Fluorescent proteins are not simple molecules, and their expressed fluorescence depends on the efficacy of their folding. Most fluorescent proteins are known to be sensitive to the oxygen and pH conditions of their production. What types of potential failure modes does this introduce into the authors' method, and can they be detected?

Together these three additions will allow a direct comparison of the authors' approach and the iGEM approach, which will be important for practitioners deciding which method to use. It is likely that the authors' method will be superior in at least some circumstances, but direct comparison is necessary for the authors to be able to make that claim.

Other significant points to be addressed during revision:

- The authors claim it is desirable to not depend on commercially supplied materials. This seems counter-intuitive, since the quality control on a commercial material is typically far higher than the quality control of execution by student trainees. Perhaps

the authors mean cost, but they provide no estimate of cost. How does this compare to the iGEM protocol cost?

- The mTagBFP2 OD700 results are quite curious. While the authors speculate on the reason, because the magnitude of the effect is similar to that of the mCherry effect, it seems problematic to draw a conclusion about mCherry without being able to more clearly determine the reason for the similar but opposite deflection with mTagBFP2.

- The authors claim significance for the bands in Figure 2C and 2D, but how is that quantified?

- Figure 3: there are quite high degrees of variation observed between the different variants of method and the variation is not systematic. Since the sample-to-sample variation is claimed to be low (though numbers are not reported and it's difficult to see on such small graphs), this implies that the variation is likely to be at the "batch" level and there may be challenges with the reproducibility of these assays.

- Figure 3C: is the fit linear or is it a linear fit on the log scale? It looks like the latter, but the caption says the former.

- Figure 3: dropping values below zero while keeping values above zero will produce a biased fit.

- Figure 4B,D: why are the units suddenly on a linear scale instead of a log scale? This makes it impossible to distinguish the low values.

- Figure 5C,D: The difference between the high and low copy vectors is potentially concerning: why does the high copy vector produce much lower expression than the low copy vector, and why is the relationship between mCherry and mTagBFP2 different between the two? Can we determine to what degree this is due to differences in the vectors vs. precision issues in the culturing vs. inaccuracies in the calibration? Presumably all share the same calibration, but there appear to be no separately prepared replicates to allow these differences to be teased apart.

- Supplementary Note 3: The calculation focuses on scattering, but what about the spectral response of silica? Could that play a role?

- Supplementary Figure 9: The standard deviations appear to be calculated from just two replicates. With such small numbers, standard deviation estimates are not particularly reliable, and the fact that behavior is not systematic across colors is concerning. There needs to be much more data here, possibly even pulling it to the main text, since quantifying the precision of these assays is critical to establishing the value of this protocol. Likewise, the relationship between buffer and conversion factor should not be gain-dependent, as shown in Supp Figure 10. I suspect that the authors are including the values below the effective sensitivity, and that this is injecting both unnecessary noise and the gain dependence in Supp Figure 10. In the iGEM protocol, these are dealt with by including a blank control, whose variation is used to establish effective sensitivity of the measurement and exclude data below that threshold. I suspect the authors could add a similar control of lysed cells with either no expressed protein or a non-fluorescent expressed protein and use it in the same manner.

- Please provide a mathematical presentation of the key elements data handling in methods or supplemental, not just a link to the R code.

Minor points:

- The authors should switch from MEFP to MESF (Molecules of Equivalent Soluble Fluorophore), which is the established nomenclature for the units they are computing. See, for example, <https://doi.org/10.1002/cyto.b.10066>

- The sentence at lines 75-77 belong in the previous paragraph, on relative units, since calibration to proxy molecules is known to be accurate when the proxy and target are well matched.
- Fit lines should be shown on all graphs where a fit is performed.
- Volume conversion of OD is not immune to cellular state, as it will be affected by the degree to which cells are clumped vs. dispersed and by any opaque cellular products.
- Figure 4B,D: Grey vs. black is harder to differentiate than the colors used in other figures. Please also include a key on the graph, like with the other figures.
- The iGEM microsphere protocol recommends units of Equivalent Particle Count (EPC) not PEMS. Was there a reason for changing the acronym?
- Figure 5C,D: Do not drop the zero data point. Use a split axis instead.

Reviewer #2 (Remarks to the Author):

This paper by Csibra & Stan proposes a novel method for the calibration of fluorescent protein expression, building on prior work in standardising fluorescence measurements so that inter-lab and inter-instrument comparisons are possible. The novelty of the work lies in the use of proteins as the calibrants themselves, rather than fluorescent molecules. This has two main advantages over existing methods: firstly, that it is possible to calibrate red and blue fluorescence in addition to green; secondly that the output values can correspond to a more meaningful measure of protein numbers per cell. This is a compelling argument and has great potential to be a meaningful tool for synthetic biology. I feel that there are a number of improvements in clarity that the paper would benefit from to make its case.

The authors postulate that this has not been attempted previously because of a lack of experience in the required techniques. Arguably the far greater challenge than doing routine protein purification, however, is reliable quantification which is a pre-requisite for valid calibration. Their results show that the microBCA and A280 assays report values with greater than 2-fold variation in some cases. This seems fairly significantly divergent, and I am unclear on why the A280 result was chosen as the 'reference' value to evaluate the accuracy of the new ECmax assay. It would seem that this variation between assays is large enough to have an impact on calibration efforts, where accuracy is critical (as discussed later on when quantifying protein numbers per cell).

It's mentioned that purified FPs are not commercially available (line 87), but various providers do sell purified GFP at least, e.g. Thermo #A42613 and Abcam #Ab84191. Have the authors considered comparing the results from these commercial products to in-house prepared proteins? This may be a necessary quality control if other groups are to use this technique, to verify protein concentration measurements are consistent or even to use directly as the calibrant for those who are only interested in GFP.

The description of the method development is a little unbalanced and could be more instructive in guiding the reader on how to perform this calibration themselves. There is a detailed description of the optimisation of two standard protein quantification techniques (microBCA and A280), after which the authors choose to use a third assay (ECmax). This makes it confusing to understand which is really necessary. Since they have discovered that the simpler method of using cell lysate and measuring ECmax provides equally good results, I feel that it should be more explicit that this is the recommended protocol rather than providing all three in the methods (line 563-573). Describing it as a 'hack' implies performing the full-length protocol with three quantification assays is still preferable, even though it doesn't appear from their results that there is any increase in accuracy for doing so. Perhaps the 'best' protocol could be included in the supplementary for completeness and the others can be linked to on protocols.io for reference.

If someone did wish to cross-validate their results by performing the longest protocol, what is the cut-off for acceptable outcomes (e.g. 2-fold variation between assays, or 3-fold, etc.)?

The second half of this paper looks at the relevance of this work for metrology in the field of synthetic biology. The main strength of the existing fluorescein calibration method is that it has been validated across a large number of different users with different instruments (ref 15), and until this has also been demonstrated for protein calibrants it should be mentioned as a limitation. In order to facilitate that type of inter-lab validation, it would be helpful to make these FP SEVA plasmids available on a platform such as Addgene, so that people can reproduce this protocol and compare their MEFP/PEMS measurements to the published results.

I appreciate that fluorescence quenching is addressed as a potential area of concern. However, the method chosen to measure this, mixing purified FPs with cell culture, does not seem to account for the effect of fluorescence being altered by FPs being inside the cell. Ref 25 looks at this so-called "inner filter effect". Can the authors comment on whether they would expect a difference in observable fluorescence between FPs in solution with cells, and FPs expressed within a cell?

The efforts to quantify the impact of red (and apparently also blue) fluorescent proteins on OD700 measurements are interesting and should be very informative for others in the field.

I recommend this paper for publication as it proposes a useful new tool, and the authors argue persuasively that the best and most relevant calibrant for fluorescence measurements would be the proteins themselves. I think it is better conceived primarily as a methods paper, and as such there should be an effort made to communicate more clearly how this method can be performed and validated by others.

Further minor comments:

- FPbase is frequently referenced but readers may not be familiar with this resource. A short description of this database and the reliability of the data would be helpful when it is first mentioned.**
- The caption for figure 2A includes information that is exclusively mentioned there, e.g. the choice of His-tag as affinity tag and expression conditions. I feel this is better suited to be moved to the main text.**
- Fig 2E shows 2 peaks corresponding to the excitation and emission spectra. It would be helpful to label them as such.**
- Fig 5C-D Can the authors suggest an explanation for why FP concentration decreases over time in the uninduced condition?**
- For fluorescein calibrations it is recommended to repeat the process and calculate new conversion values regularly to account for instrument drift. Presumably this is also true with FP calibrants. If so, is it possible to use the same purified protein after a period of months, or would they be expected to degrade in storage over time?**
- Supplementary figure 1B is missing labels for the FPs.**

REVIEWER COMMENTS

Reviewer #1 (Remarks to the Author):

The authors of this manuscript propose an alternative to the iGEM plate reader calibration assay. In their protocol, the authors exchange small-molecule fluorescent dyes for fluorescent proteins as a calibrant. Fluorescent dyes with a well-matched spectrum already provide a usefully accurate estimate of fluorescent protein molecule count. By switching the calibrant from a proxy molecule to the actual molecule of interest, however, the authors' assay has the potential to allow an increase of the accuracy of estimates, which will in turn increase the comparability and reusability of data. The authors approach, like the iGEM approach, is clearly superior to normalization against a "standard" strain, in this case because the lysing of cells and use of absorption to quantify protein count mitigates most aspects of cell culture variation (but not all, per below).

The authors provide strong evidence that their protocol is indeed able to provide reasonably accurate estimates of fluorescent protein count, as well as making use of this approach to evaluate the magnitude of impact that cell density and protein absorbance have on protein count estimates. It is not yet clear, however, whether the iGEM approach or the authors' approach has better efficacy.

We thank the reviewer for their considered and helpful response.

Taking into account all of reviewer 1's comments, we have included in total 1 extra figure (Supplementary Fig. 11), 2 extra tables (Supplementary Tables 7-8) and 2 extra supplementary notes (Supplementary Notes 3 and 6), as well as extended 2 existing figures (Figure 3, Supplementary Fig. 9), detailed in the responses below. We have also endeavoured to produce detailed responses to all of the reviewer's following points. These summarise our observations obtained from investigating the precision of our FP-based calibration method (purified and non-purified versions) by measuring the variability of results from independent replications, as well as our experiences with the use of commercial FPs as calibrants, among other comparisons with the use of fluorescein vs FPs.

We note that our FP-based method is not intended to replace the iGEM method i.e. calibration using fluorescein, it is meant to be complementary – a different approach for a slightly different end. The serial dilution of powdered or pre-dissolved small molecule fluorophores will always constitute a simpler process than one that requires the production of a calibrant first. Clearly, the method we describe cannot compete on simplicity. However, the major point is that small molecule calibrants such as fluorescein can standardise fluorescence values only to units of (those same) small molecules, i.e. 'molecules of equivalent fluorescein' – not to units of protein molecules as the reviewer states above – and in many cases, protein molecule numbers will be essential in understanding how to build effective genetic circuits, as the order of magnitude and relative abundance of host and synthetic proteins will be important.

Further, the use of FPs as calibrants simplifies the process of finding approximate spectrally matching small molecules, which allows a wider range (indeed, any) FPs to be calibrated, even blues, yellows and far reds. Additionally, the ability to count (for instance) BFP, GFP and RFP in the same cell and therefore meaningfully compare protein levels *between* proteins in the *same* genetic circuit is also enabled by this process. It is these outcomes that we anticipate biologists (perhaps especially synthetic biologists) will find useful in our method.

Overall, this is interesting work and deserving of publication once a few key issues are addressed, these being:

1) Lack of a head-to-head comparison between the iGEM methods the authors cite and their own methods. While the authors are aware of these assays and claim an increase in accuracy over the iGEM methods, they do not actually demonstrate this. In particular, a comparison with flow cytometry data would be valuable, in order to determine the actual accuracy of the fluorescence/cell estimates produced by the authors. Calibration beads with known quantities of GFP and mCherry are both commercially available, and can be used in the same manner as the MEFL-calibrated beads in the iGEM studies.

We thank the reviewer for this comment. We do not claim increased accuracy of our method over a method using small molecule fluorophores (see our comment above): we claim that calibration of FP signal to the same (purified) FP allows us to convert arbitrary units to units of protein.

As this comment was similar to one made by reviewer 2 about the comparison with commercially purified FPs, we will start with a discussion on those (Point 1a) and address flow calibrants (Point 1b) below this.

1a. Commercially purified FPs.

While we too initially thought that commercially available FPs might be a good way to validate our methods, unfortunately we no longer believe that these invariably have higher reliability or accuracy than purified FPs that we can produce in house with our published protocols. This is due to a number of reasons, including the fact that

most GFPs on the market are very different spectrally to 'modern' GFPs and that there is a lack of understanding as well as a lack of detailed quality controls (QCs) by suppliers selling these proteins. However, as the suggestion to use commercial FPs seems to be a common one, we have set out our reasoning explicitly and in detail in our newly included Supplementary Note 3 (reproduced for convenience below).

Despite these points, we identified a few commercially available GFPs that may function reasonably well as calibrants for users who cannot produce their own. We ordered two, one EGFP and one TurboGFP, but unfortunately the EGFP did not arrive in time to test it. The TurboGFP (Pierce Recombinant GFP Protein #88899) was tested with two independent calibrations, described in the new Supplementary Note 3, and presented in the new Supplementary Table 8. From our analysis, we observed that good quality commercial GFPs can provide similar results to in-house produced calibrants.

Paragraph added in the main text:

"The use of commercial fluorescent proteins as calibrants

We were interested in testing whether our purified FPs gave similar RFU to molecule conversion factors as commercially available FPs. Very few FPs are available commercially, and the majority are green FPs, so we focussed on those. We compiled a table of available GFPs to ascertain the best candidates to order (Supplementary Table 7). Surprisingly, many were based on first generation GFPs that preferentially excite in the UV range (unlike GFPmut3 or sfGFP), had incomplete datasheets lacking protein sequence, brightness or spectral information, or were not subject to explicit quality controls (Supplementary Note 3). This suggested their production may be less rigorous than the methods described in this paper and makes them difficult to recommend as calibrants (Supplementary Note 3). Nonetheless, using a commercial TurboGFP, we obtained a relative conversion factor of 95.0% compared to the value obtained using our purified mGFPmut3, which suggests that FPs from in-house and (carefully selected) commercial sources may be used interchangeably (Supplementary Table 8)."

Text added in the supplementary material:

"Supplementary Note 3. The use of commercial fluorescent proteins as calibrants

As described in the main text, we were interested in testing whether our purified FPs gave similar conversion factors as commercially available FPs in order to validate our methods and results, with a focus on GFPs. The result of our efforts to compare commercially available GFPs is provided in Supplementary Table 7.

Unfortunately, we observed several problems with the use of these products as reliable calibrants. First, the majority of the available FPs are either the original "wild-type" *Aequorea victoria* GFP (FPbase: avGFP/1XF1B) – which is almost never used in modern cell biology – avGFP-related but obscure proteins such as alphaGFP (FPbase: B28N7) or Q80R (FPbase 799YV), or commercially-developed proteins such as 'GFPSpark' (not on FPbase). Some of these proteins do not have FPbase entries or recorded fluorescence spectra, and others, such as avGFP, have significantly different spectra from the 'modern' GFPs like GFPmut3 or sfGFP, absorbing light maximally in the UV range, rather than the blue (488 nm) range. We also found that some FPs were poorly annotated in their datasheets in a way that suggested a lack of quality control (QC; see Supplementary Table 7), and technical support staff occasionally described avGFP as 'normal' or 'standard' GFP – suggesting an erroneous perception that such GFPs are widely used, or that the identity of the GFP in question is not considered important for most applications. Some suppliers confirmed they do not test their FP batches for fluorescence or carry out QC on their fluorescence activity.

These observations suggested that we would need to validate commercial FPs against our FPs, rather than the other way round. We selected the TurboGFP available from Pierce for testing and performed calibrations in T5N15pi buffer with the ECmax assay (Supplementary Table 8). Our results confirm that this FP produces similar results to our mGFPmut3 calibrants, with a mean conversion factor of 95.0 % compared to the conversion factor of mGFPmut3. This suggests that users can use commercial FPs to gain approximate conversion factor estimates provided they check for a good match between the properties of their FPs and those available.

Of interest is the fact that in the few weeks since we ordered the Pierce TurboGFP, this product has already been discontinued. For this and the above reasons, we would generally recommend that users make their own calibrants. These are easy to produce, bespoke to the specific FP used in users' own cells and genetic constructs and can be affordably produced in high quantities. Moreover, if users purify FPs and run an SDS-PAGE to ascertain purity, they will have done as much QC as is typically done on commercially purified FPs. As our protocol involves taking an entire absorbance spectrum, this can act as an added QC step to verify FP quality against expected spectra on FPbase. Dilution in protease-inhibitor free buffer and the use of UVclear plates allows users to also check the protein's concentration using two different methods, providing yet another QC step over a commercial protein."

Our new Supplementary Table 8 illustrates the relative conversion factors obtained from the mean of two independent repeats of a calibration experiment using TurboGFP (the stock tube contained only enough for two repeats), compared to the mean conversion factor obtained from in-house produced purified mGFPmut3.

The new Supplementary Tables 7 and 8 are provided as attached files.

Supplementary Table 7: Comparison of commercially available GFPs. The properties of purified GFPs available as of May 2022 listed in order of suitability as calibrants for calibration of the mGFPmut3 protein used in this work. The left of the table compares the GFPs in terms of their identity with FPbase-recorded protein sequences and fluorescence properties, and the right in terms of the purification and QC procedures of the supplied proteins.

Supplementary Table 8: Relative conversion factors. Conversion factors (relative fluorescence units/molecule number) from purified, sonicated lysate and chemical lysate calibrants are all shown as relative to the mean of the purified equivalent fluorescent proteins (purified mCherry values were compared to the mean of all purified mCherry values, and sonicated mCherry lysate values were also compared to the mean of purified mCherry values, etc). One anomalous value (mTagBFP2 purification set 1) was removed from the calculation of the mean as described in the text. TurboGFP is shown as compared to the purified mGFPmut3 samples. Sets here refer to independent replicates. Abbreviations: stdev, standard deviation; CV, coefficient of variation.

1b. Commercially available FP-linked flow beads.

We thank the reviewer for the suggestion to compare our results with those obtained with GFP-linked flow cytometry beads, but after careful consideration, we believe that the comparison with flow cytometry is beyond the scope of this paper. Our plate reader calibration protocols are similar to the iGEM methods in that we undertake two calibrations – one for fluorescence and another for cell number – and we obtain ‘molecules per cell’ by dividing the molecule counts from the fluorescence calibration by our cell counts from our cell number calibration. Our fluorescence calibration differs from the iGEM protocols in that it uses FPs, but our cell number calibrations are identical – we use microspheres, which were found by Beal et al., 2020 to give the most reproducible cell number calibrations.

The reviewer suggests using flow cytometry to validate ‘per cell’ counts using plate readers, however, if identical calibrants (e.g. fluorescein in solution and fluorescein on beads, or GFP in solution and GFP on beads) are used to calibrate fluorescence, then the only thing that a flow cytometry to plate reader calibration compares is the quality of the cell number calibration using microspheres. As our protocol uses previously published microsphere calibration protocols, and this comparison has been completed before (Fig. 5 of Beal et al., 2020 or Fig. 3 of Fedorec et al., 2020), we don’t feel that carrying out such a comparison would add to this paper, whose focus is the use of in-house produced FP calibrants.

2) Quantification of precision: the authors make many qualitative statements about the efficacy of their assay, but never provide any quantification of the precision of their approach across independently prepared calibration curves. While an interlaboratory study like the iGEM study would be ideal, that is clearly out of scope for the current manuscript (though it would be good to see it discussed as potential future work). The authors must, however, provide numerical quantification of the precision of their results across multiple independently prepared replicates, in order to provide a direct numerical comparison of the precision of the authors’ method and the iGEM method.

We thank the reviewer for this comment. We have extended our analysis of the precision of the technique based on this comment.

2a. Precision of purified calibrants

The original manuscript did contain a figure (Supplementary Fig. 9) that compared the conversion factors obtained from two independent replicates of all three proteins across three protein assays and two buffers. However, this is a very large experiment to replicate numerous times, and unnecessary after we had selected the best method (ECmax) and the best buffer (T5N15pi). So, for this precision analysis, we focussed on repeating calibrations using the T5N15pi buffer and the ECmax assay, for all three proteins. We repeated the purifications and calibrations in two more independent repeats, for which the results are shown below as relative conversion factors, compared to the first purified batch (called set 1).

Rebuttal Figure 1. Comparison of conversion factor prediction using purified FPs across four independent repeats, using the ECmax assay and T5N15pi buffer. Conversion factor (cf) predictions were compared across independent repeats by calculating the fold difference between the cf of the current set against the cf of set 1. Plots display mean fold differences across the gains and error bars represent the standard deviations.

Overall, we found the protocol reproduced well, as repeated sets of purifications for mCherry and mGFPmut3 produced conversion factors that differed by under 20%. The replications did however bring to our notice that for mTagBFP2, in which we had originally observed a large difference of 1.6-fold between purification sets 1 and 2 (see original Supplementary Fig. 9B), we continued to observe a large discrepancy between set 1 and the other three sets (average fold change, 1.67-fold), indicating that mTagBFP2 is not inherently variable, but rather our first result is likely to be an outlier.

For this reason we have decided to present the updated Supplementary Fig. 9B and 9C as conversion factor values in comparison with set 2, i.e. SF9B shows values of conversion factors obtained in the first set of purifications divided by the conversion factor obtained in the second set. Similarly, a new figure, SF9C, shows the values obtained in sets 1, 3 and 4 compared to those in set 2.

The above data represented in comparison with set 2 is shown below:

Rebuttal Figure 2. Comparison of conversion factor prediction using purified FPs across four independent repeats, using the ECmax assay and T5N15pi buffer. Conversion factor (cf) predictions were compared across independent repeats by calculating the fold difference between the cf of the current set against

the cf of set 2. Plots display mean fold differences across the gains and error bars represent the standard deviations.

To address this point, we have included the following in the main text:

“Robustness of calibration protocols using purified FPs

To further investigate the reproducibility of this calibration method, we completed two more independent repeats of calibrations with all three FPs using the T5N15pi buffer and the ECmax assay. From this data, we observed that one of our calibration runs obtained with mTagBFP2 in our original experiments (Supplementary Fig. 9A, mTagBFP2 set1) produced an anomalous value for the mTagBFP2 conversion factor, which was 1.67-fold higher than the other replicates. As a result, we present a comparison of the reproducibility of conversion factors as compared with conversion factors from set 2 of the original data (Supplementary Fig. 9C). The full data is provided in Supplementary Table 8. Our data suggests that, generally, the conversion factor values obtained using the described method are highly reproducible (they differ by less than 20% in all cases except for the anomalous mTagBFP2 value). Therefore, we recommend users conduct two independent calibrations for each FP, and exercise caution if the replicates differ by over 20%.”

The complete new Supplementary Fig. 9 is presented below:

Protocol robustness with purified FPs:

Supplementary Figure 9. Effect of protein assay and buffer on concentration and conversion factor estimation of three FPs. A. Full data from the systematic comparison of three FPs across two buffers, three assays and two purifications. Plots show the measured concentrations of each FP dilution (FPs in rows) across two purifications (set1 and set2) with the three assays (microBCA in blue, A280 in black, ECmax in red), across two buffers (open circles, T5N15; crosses, T5N15 with protease inhibitors). Each serial dilution was a 2-fold dilution, where 11 dilutions were prepared and measured in duplicate. Points represent the mean of the duplicate values. All 11 dilutions were measured with the A280 and ECmax assays but only the top 8 dilutions were tested with the microBCA assay. The values for the A280 and ECmax assays were normalised for scatter as indicated in Supplementary Table 2. Any missing data points had concentrations recorded as being below 0.01 ng/μl. **B. Comparison of conversion factor prediction across batches.** Conversion factors (cf) across batches (set1 vs set2) were compared by calculating the fold difference at each gain. Plots display mean fold differences across the gains and error bars represent the standard deviations. Colours and shapes are identical to (A). **C. Comparison of conversion factor prediction across further independent repeats using ECmax assay and T5N15pi buffer.** Showing conversion factors across the two existing sets of purifications in A-B (sets1-2), plus two further sets (sets 3-4) compared to conversion factor of set 2 (*left*) or the average conversion factor (*right*). Note that the set 1 value for mTagBFP2 was excluded from the calculation of the average cf (*right*).

The full data, detailed in Supplementary Table 8, is supplied as an attached file.

2b. Precision of purified calibrants

As we now know that FPs do not require purification to act as usable calibrants, we also repeated our lysate-based calibration method for all three FPs, over three more independent repeats for the sonicated samples. (The chemical lysis method was also repeated.) As before, we compared the relative conversion factors obtained from lysates with those obtained from purified calibrants. We used updated values for the comparator – the mean of the conversion factors obtained from purified FPs – to include all the new data and exclude the anomalous mTagBFP2 value. Having done this, we observed that the additional independent repeats have clarified that the quality of the data obtainable from lysates is high, even for mTagBFP2 (where the mean conversion factor obtained using lysates was 95 % that of purified proteins). Originally, the data for mTagBFP2 appeared less reliable due to the large skew in the mean data point due to the anomalous result in our first purification.

We added this data to the existing Fig. 3F and expanded the figure to display the sonicated and chemical lysis values separately to avoid over-plotting. We have kept the concentration quantification data from set 1 (from the original Fig. 3E) but have included concentration quantifications from all sets of sonicated lysates in the supplementary material (new Supplementary Fig. 11A-B). The full data set has been added to Supplementary Table 8.

We have also included a figure on the stability of the calibrants in lysate, showing reasonable stability for a number of weeks of storage of pre-prepared calibrants (Supplementary Fig. 11C).

We have modified the relevant section of the main text as follows:

“Putting these through an ECmax assay and fluorescence assay, we observed it was possible to quantify FPs in crude lysates with high sensitivity (to 1 ng/μl; Supplementary Fig. 11A), and to obtain almost identical conversion factor values as from our purified FPs, using the mean conversion factor from all (non-anomalous) purifications used as a comparator (Fig. 3E-F, Supplementary Fig. 11B, Supplementary Table 8). Of the two lysis methods tested – sonication and chemical lysis – the former produced more accurate results (conversion factors were closer to the expected conversion factor from purified proteins – 90-99% of expected values) and they also had high precision (low variability, with coefficients of variation between 0.02-0.12, similar to the CVs observed with purified calibrants). Repeated testing of calibrants prepared by sonication suggests that FPs maintain stability in lysates when stored at 4 °C for a number of weeks (Supplementary Fig. 11C). Using the ECmax assay for FP quantification, it is thus possible to remove the purification step altogether without compromising calibration accuracy and precision.”

The new Fig. 3D-F:

Figure 3. Systematic characterisation of calibration protocols reveals novel assay that allows calibration in crude lysates. D-F. FP calibration in crude lysates. D. Sample preparation and assay workflow. Calibration in lysate requires fewer steps: affinity purification is omitted, buffer exchange is not required (optionally, lysates may be concentrated to increase signal), and dilution series are subject to only two assays requiring no commercial reagents or incubation steps. **E. Measured protein concentrations of FPs in lysate with ECmax assay.** Dilution series using lysates obtained by sonication (orange) or chemical lysis using the commercial B-PER reagent (Thermo Fisher, green), were measured with an absorbance scan and `get_conc_ecmax()`. Data plotted as in B. The top dilution of each series using chemical lysis was removed due to excessive sample scatter. The full data set can be found in Supplementary Fig. 11A. **F. Difference between conversion factors using lysate vs purified protein.** Conversion factors were compared by calculating the fold difference (lysate/purified protein) at each gain. Plots display mean fold differences across the gains and error bars represent the standard deviations. The full data set can be found in Supplementary Fig. 11B.

The new Supplementary Fig. 11:

Supplementary Figure 11. Performance of FPs in lysate as calibrants. A-B. Full data from independent repeats of the lysate protocols shown in Fig. 3E-F. Plots show the measured concentrations (A) and relative conversion factors compared to purified FPs (B), across up to 4 sets of experiments using lysis by sonication (orange) or chemical lysis (green) using the ECmax assay and T50N300+pi buffer. **C. Stability of two sets of sonicated lysates over a period of storage at 4°C.** Two of the batches of calibrants prepared for A-B were stored at 4°C and remeasured over a period of six weeks. Plots show their measured conversion factors relative to the conversion factors measured at 0 weeks.

3) Potential impact from using a biological calibrant as opposed to an independent calibrant. Fluorescent proteins are not simple molecules, and their expressed fluorescence depends on the efficacy of their folding. Most fluorescent proteins are known to be sensitive to the oxygen and pH conditions of their production. What types of potential failure modes does this introduce into the authors' method, and can they be detected?

We thank the reviewer for adding these valid points.

3a. pH and temperature

We take care to keep the pH of the calibrant buffer similar to that of cells (ours is pH 7.5) and the temperature of calibrations is also kept at the temperature used for *E. coli* experiments (30 °C). We had originally included a discussion of the point about pH in a previous version of our manuscript, but it was edited out for the sake of the word count.

3b. Oxygen

Regarding oxygen, the point is also valid. Fluorescent proteins require oxygen to mature. The major point for users to remember is that this means that newly-expressed FPs take time to become fluorescent. However, in well-aerated cultures such as flask cultures, there is no shortage of oxygen for this maturation process, and in

any case, the process of cell harvesting and lysis allows time for protein maturation to take place. We originally allowed for maturation by producing FP calibrants on one day and allowing them to mature overnight at 4 °C. However, in later experiments we have been doing the calibrations on the same day as calibrant preparation, and have not observed a change in measured conversion factors – in other words, the protocol itself allows time for such maturation.

3c. Folding

It is certainly true also that fluorescent proteins require beta-barrel formation (correct folding) for their fluorescence. However, the major cause of FP misfolding in cells is their improper fusion to other proteins that are unable to tolerate such fusions, causing both protein domains in the fusion protein to misfold. This is not an issue for FP calibrants since these are not fused to other proteins.

We have added a clarifying note about the conditions for FP production and testing in the discussion:

“While fluorescent proteins are biological molecules whose fluorescence activity is dependent on their condition of production and testing (such as pH and the availability of oxygen), the use of a few well-established techniques from standard bacterial protein overexpression protocols allows such problems to be avoided. For instance, the expression protocol we describe is designed to produce high levels of protein: we use high copy vectors and long expression times. However, we grow cells at low temperatures of 25-30 °C for the expression, which minimises the chance that overexpressed proteins will misfold or aggregate (Green and Sambrook, 2001). The presence of aggregates can be checked by SDS-PAGE after the clarification step that separates the soluble and insoluble fraction (Green and Sambrook, 2001). We rarely see significant aggregation (Fig. 2C). Oxygen is also required for chromophore maturation (Shaner et al., 2005), however shaken flask cultures will be well aerated, and the lysis and/or purification procedure gives time for any remaining immature proteins to mature during the protocol itself. (Simply exposing cells to air allows maturation of FPs expressed in anoxic conditions (Zhang 2005).) Furthermore, the buffer used for the calibration assays should match the cellular pH (we use pH 7.5). We also conduct calibration assays at a temperature that matches our bacterial assays (though we do not expect that minor temperature changes would have a significant effect on FP brightness or behaviour).”

We have also added a paragraph on matching *in vitro* and bacterial pH:

“Certain sources of error cannot be adequately addressed by calibration alone. While some have noted that pH can affect the molecular brightness of certain FPs (Hirst *et al.*, 2015), this could not be compensated for analytically without user input detailing both the pH response profile of the included FPs and the pH of their cells. Fortunately, since the cellular pH in *E. coli* is limited between pH 7.2-7.8 (Wilks and Slonczewski, 2007), and even pH-sensitive FPs exhibit only mild (<10%) variation in molecular brightness for pH values between 7-8 (Kneen *et al.*, 1998; Roberts *et al.*, 2016), pH-dependent changes in molecular brightness are unlikely to have a large effect on quantifications.”

3d. Small molecule fluorophores

It is interesting that condition-specific fluorescence is often invoked against the use of fluorescent proteins as accurate calibrants, but this is rarely remarked on for small molecule fluorophores. In fact, the fluorescence of small molecules is also sensitive to conditions, particularly pH and temperature. Fluorescein in particular is well known to be highly pH sensitive (see for example Le Guern *et al.*, 2020 (doi: 10.3390/ijms21239217)), and yet this is the calibrant proposed in the iGEM protocols. In these protocols, it is recommended that fluorescein is resuspended in PBS, which is typically a near-neutral pH, at which its fluorescence is certainly lower than in its usual solvent, 0.1M NaOH (pH 13).

Together these three additions will allow a direct comparison of the authors' approach and the iGEM approach, which will be important for practitioners deciding which method to use. It is likely that the authors' method will be superior in at least some circumstances, but direct comparison is necessary for the authors to be able to make that claim.

Other significant points to be addressed during revision:

- The authors claim it is desirable to not depend on commercially supplied materials. This seems counter-intuitive, since the quality control on a commercial material is typically far higher than the quality control of execution by student trainees. Perhaps the authors mean cost, but they provide no estimate of cost. How does this compare to the iGEM protocol cost?

We thank the reviewer for allowing us to clarify this point.

We did not claim that it was desirable not to depend on commercial materials; rather, we said that commercially available purified versions of all FPs are not generally available. If they were, the protocol could of course be much simpler, and similar to the iGEM method. However, it seems additionally true that many, if not most, commercial FPs are less trustworthy than those made in our own hands, and subject to less QC (as discussed extensively above).

While we agree that it is admirable if a protocol is so accessible that even students with no previous laboratory experience can obtain good results, and we have worked to make this protocol as simple as possible, we note that many of the beneficiaries of our protocol will be experienced students and postdocs, for whom a clear protocol, an SDS-PAGE and a few absorbance scans (totalling more QC than the average commercial FP) will be trivial.

We have not attempted to compare the protocols on the basis of cost. Commercial GFPs cost £40 on average per 10 µg required for a single calibration, so would cost £80 for two independent repeats (Supplementary Table 7). The cost of the sonicated lysate will consist of media, inducer, standard buffer components, lysozyme, protease inhibitors and optionally an Amicon column – which we have estimated at £10.18 per preparation. Our typical yields from the sonicated lysate protocol are on the order of 100 µg protein from 20-40 % volume of a 50 ml culture.

In order to summarise the comparison between the fluorescein and FP protocols, we have included an extra paragraph in the discussion:

“Current users of fluorescein calibrations may be interested in how this method compares with calibration using a serial dilution of fluorescein. The protocol for fluorescein calibration is certainly cheaper (£0.23 per calibration using details from Beal et al., 2018) owing to the fact that fluorescein is a low-cost fluorophore, and simpler, as the use of commercial calibrants obviates the need for calibrant preparation or concentration determination. The key advantage of FP calibration lies in its ability to be used to directly convert arbitrary units directly to molecules of one’s desired FP, coupled with the freedom to produce bespoke calibrants from any desired FP, or indeed multiple FPs, from across the spectrum, whether or not they match the properties of fluorescein. While commercial FPs cost an average of £39.05 per calibration (assuming use of 10 µg FP per calibration, Supplementary Table 7), the estimated cost of producing one batch of FP calibrants in lysate is only £10.18, from which the standard yields are approximately 100 µg FP (using only a fraction of the culture). While FPs can display condition-dependent fluorescence, as discussed above, these effects can be minimised: indeed, they can be used to test the temperature or pH sensitivity of a user’s FP under controlled conditions. Furthermore, fluorescein and other small molecules also display condition-specific fluorescence, though this point is rarely considered (Le Guern et al., 2020). While it is difficult to directly compare the precision of this method with published data on fluorescein, the FPCountR protocol is clearly highly reproducible. Calculated conversion factors are typically within $\pm 10\%$ of the expected value and coefficients of variation between independent replicates are typically below 0.1 (Supplementary Table 8). Following on from the exemplary work on fluorescein (Beal et al., 2018), there is clear potential for future inter-lab studies to extend our understanding about the accessibility and reproducibility of the FPCountR protocol.”

- The mTagBFP2 OD700 results are quite curious. While the authors speculate on the reason, because the magnitude of the effect is similar to that of the mCherry effect, it seems problematic to draw a conclusion about mCherry without being able to more clearly determine the reason for the similar but opposite deflection with mTagBFP2.

The mCherry effect is a documented effect seen by others, and the reasoning based on absorbance in the 600 nm range is compelling (Hecht et al., 2016).

We note that we have replotted the data with the updated conversion factors for Fig.6. As the conversion factor for mTagBFP2 has been revised down by ~1.5-fold, correspondingly the molecule estimates have increased the same factor. This means we observe a ~5% error in OD600 estimates for mCherry at about 250,000 molecules per cell, whereas for mTagBFP2, we observe a requirement for about 450,000 molecules before a similar error is measured in the opposing direction. Together with the fact that mTagBFP2 does not absorb in this region (results from our studies and the literature, Hecht et al., 2016), we think this suggests that the source of this effect is different than what is happening with mCherry overexpression. We hope that if others have similar observations in the future, the availability of our data on this effect will allow the testing of suggested causes, which may be related to the behaviour of the FP or the *E. coli*.

- The authors claim significance for the bands in Figure 2C and 2D, but how is that quantified?

The assessment of protein yield and purity was qualitative, not quantitative. This sort of analysis is routine in protein purification and our comments in the text compared the results obtained with this FP to the authors' experience of purifying other proteins. For instance, some proteins express poorly. These can be identified easily by qualitative analysis on SDS-PAGE by virtue of not being visible on a gel that separates the total protein content of a bacterial culture (as only the most abundant proteins will be visible on such a gel). In contrast, Fig. 2C-D show that the band for mCherry is visible, which shows that it is one of the most abundant proteins in those cells. Secondly, many proteins are difficult to work with and purify. These can be identified on SDS-PAGE: they are often present in higher amounts in the insoluble fraction than the soluble fraction, suggesting they misfold and/or aggregate. In contrast, Fig. 2C shows FPs are present mostly in the soluble fraction. Poor protein purification protocols can also be identified by looking at a PAGE gel of the fractions from each step of a purification: flowthrough lanes (proteins that did not remain attached to the affinity matrix during the binding step) may contain visible levels of the desired protein, and elution lanes may contain high levels of contaminating proteins. In contrast, Fig. 2D shows that this is a very 'clean' purification: flowthrough lanes F1 and F5 show no overt contamination with FPs, whereas elution lanes are clean of other bands.

- Figure 3: there are quite high degrees of variation observed between the different variants of method and the variation is not systematic. Since the sample-to-sample variation is claimed to be low (though numbers are not reported and it's difficult to see on such small graphs), this implies that the variation is likely to be at the "batch" level and there may be challenges with the reproducibility of these assays.

The purpose of the experiment in Fig. 3A-C was to test the performance of each protein concentration assay (microBCA, A280 and ECmax) for its ability to produce a decent linear range from which we could obtain reliable concentration estimates, and then to compare the resultant conversion factors using the ratio between the fluorescence assay (for RFU) and concentration assay (for molecule number). We stated the number of experiments in all figures in the figure legends: we tested all three FPs across 3 methods and 2 buffer conditions, each repeated twice independently. The full results in Supplementary Fig. 9A illustrate this.

We certainly observed differences between the assays, particularly in the concentrations that could be detected reliably as judged by the linear range, and in their tolerance to the buffer components, as shown in Fig. 3B. As described in the text, the ECmax assay generally performed better than the others in this regard. We certainly agree that not all the methods produce the same conversion factors, as shown in Fig. 3C. The microBCA assay appears to lead to an underestimation of concentration, leading to an overestimation of the conversion factor compared to the A80 assay by almost 2-fold.

As described above, we have conducted several extra independent repeats using both purified and lysate calibrants and found their reproducibility to be very good (CV < 0.1; Supplementary Table 8).

- Figure 3C: is the fit linear or is it a linear fit on the log scale? It looks like the latter, but the caption says the former.

We are not sure which figure the reviewer is referring to here, since Fig. 3C does not include a fit. If the reviewer is referring to Fig. 3B, these figures represent the dilution series relative concentrations (x axis) and the concentration measured by the relevant assay (y axis). We have removed the fit lines to make the plot look less busy. But yes, the relevant R function fits a linear model between dilution and concentration.

- Figure 3: dropping values below zero while keeping values above zero will produce a biased fit.

We are not sure which figure the reviewer is referring to here, since Fig. 3 does not refer to dropping values below zero.

- Figure 4B,D: why are the units suddenly on a linear scale instead of a log scale? This makes it impossible to distinguish the low values.

We used a linear scale for Fig. 4 to allow for a better appreciation for the differences observed in the normalised values. We have included insets for low value plots for clarity.

- Figure 5C,D: The difference between the high and low copy vectors is potentially concerning: why does the high copy vector produce much lower expression than the low copy vector, and why is the relationship between mCherry and mTagBFP2 different between the two? Can we determine to what degree this is due to differences in the vectors vs. precision issues in the culturing vs. inaccuracies in the calibration? Presumably all share the same calibration, but there appear to be no separately prepared replicates to allow these differences to be teased apart.

We have investigated this phenomenon further by conducting two more independent replicates of each experiment presented in Fig. 5CD: a titration of arabinose using the high copy mCherry, low copy mCherry, high copy mTagBFP2 and low copy mTagBFP2 vectors. The results are presented below:

Rebuttal Figure 3. Comparison of independent repeats of arabinose titrations with mCherry and mTagBFP2 vectors. mCherry (*top*) and mTagBFP2 (*bottom*) expression from medium (pS361, p15A) and high (pS381, colE1) copy vectors was induced at a range of arabinose concentrations and quantified in molecules per cell with a calibrated plate reader at 420 min post induction. Displayed points show the mean values from each experiment, which consisted of three biological replicates. Error bars indicate standard deviations.

The above figure compares the data across the three experiments at identical time points, showing the mean and standard deviations across three biological replicates for each experiment. We observed that the repeats (experiments #2 and #3) gave similar values to one another, but generally lower values than the initial experiment presented in the original Fig. 5C (shown in black). The differences in the mTagBFP2 values between experiments 1 and 2/3 were mild and perhaps in line with biological and technical variations expected from experiments conducted several months apart. However, the discrepancies seen between mCherry experiments were larger and suggest that our initial experiment was unrepresentative of the general behaviour of our mCherry vectors. Related to this, we additionally observed that the high copy (pS381) mCherry expression vector, grown in the presence of 0% arabinose, did not produce the leaky expression phenotype observed during the first experiment:

Rebuttal Fig. 4. Comparison of independent repeats of arabinose titrations with pS381_ara_mCherry. mCherry expression from a high (pS381, colE1) copy vector was induced at a range of arabinose concentrations and quantified in a timecourse assay as molecules per cell in a calibrated plate reader. Data from each of three independent experiments (1,2,3) is presented side by side. Data was collected from three biological replicates, each of which is plotted.

We suspect therefore, that the discrepancies between the mCherry molecules per cell across the repeats is largely due to the fact that in the first experiment we observed a high level of leaky expression in our overnight cultures (in the order of 1000s molecules per cell at time point 0, when arabinose was added), that inflated the total mCherry values throughout the time course in experiment 1. The other experiments show lower overall mCherry values at all timepoints due to mild to undetectable levels of leakage (mCherry levels at 0% arabinose (in red) are often so low as to not be plottable even on an extended y axis starting at 1 molecule/cell).

We therefore decided to replot the data in Fig. 5C-D using data from mCherry experiment #2 for the top panel, and averages from experiments 2 and 3 in the bottom panels, as the newer data seem to be more representative of the behaviour of these plasmids. In addition, as our calculated average conversion factors for each FP have been updated with new data since our original manuscript, particularly for mTagBFP2, we have reprocessed the data with the new conversion factors to take into account these changes. The resultant new Fig. 5CD is as follows:

Figure 5. Absolute quantification of *E. coli* timeseries data in molecules per cell.

C-D. Absolute protein quantification in molecules per cell (C) and molar concentration (D). mCherry expression (*top*) from medium (pS361, p15A) and high (pS381, colE1) copy vectors was induced at a range of arabinose concentrations and quantified in a timecourse assay in a calibrated plate reader. Data was processed as described in (A) and cell estimates based on microsphere calibrations were used to calculate per cell values (C), or OD-specific cell volume data from Volkmer et al., 2011 was used to calculate molar concentrations (D). Data was collected from three biological replicates, each of which is plotted. (*Bottom*) mCherry expression from the top panel is compared with mTagBFP2 expression from an identical assay, plotted against arabinose concentration at 420 min post induction. Displayed points show the mean values from two independent experiments, each of which tested three biological replicates. Error bars indicate standard deviations.

In order to bring the text in line with the new data, we have edited the following sentences:

“Using these vectors, we obtain figures for mCherry abundance that vary between about 900 to 70,000 molecules per cell for p15A, and 200 to 200,000 for colE1.”

“Interestingly, measurements from identical SEVA vectors revealed that while the FP abundances were in the same order of magnitude, mTagBFP2 accumulated to higher levels per cell than mCherry by 3.8-fold on average, despite sharing the same promoter, 5' untranslated region, ribosome binding site and N-terminal protein sequences (Fig. 5C-D).”

We have also removed the sentence: “Leaky expression could be quantified in the colE1 vector as contributing hundreds of mCherry molecules per cell even without induction, whereas in the p15A vector it was merely ten copies per cell.”

To answer the reviewer’s specific question about mCherry vs mTagBFP2 plasmids, in which we observe higher levels of mTagBFP2 than mCherry in molecules per cell, the straightforward explanation is that the mTagBFP2 construct is produced at a higher rate than the mCherry construct, due to translation initiation effects beyond the first few codons that were standardised, or to the possibility that the mTagBFP2 protein is degraded more slowly, leading to higher overall accumulation. We have added this suggestion to the text:

“This could be due to translation rate effects from RNA level effects in the coding sequences beyond the first 11 standardised codons, or else due to differences in degradation kinetics between the two proteins.”

To answer the reviewer’s specific question about the behaviour of the high copy mCherry vector, our experimental results across all repeats were consistent: at high arabinose concentrations, mCherry levels from our colE1 plasmid are similar to those of mTagBFP2 (~200,000 molecules/cell), and higher than mCherry levels from our p15A plasmid (~70,000 molecules/cell). However, at low arabinose concentrations (over 1000-fold lower than saturation) mCherry levels from the colE1 plasmid reproducibly result in lower mCherry molecules per cell than from the p15A plasmid. This suggests a reproducible effect.

We cannot currently explain these results, but we suspect that the differences must be due to genetic factors to do with the presence of the mCherry expression construct. We do not believe this is due to precision issues in the calibration, as we have not observed differences in the precision of calibration between different proteins. The culturing conditions were also identical in experiments #2 and #3 between the high-copy mCherry and high-copy

mTagBFP2 vectors, since they were grown side by side in the same plate for these experiments. We note that the purpose of these experiments was not to do a deep investigation of the behaviour of these specific vectors per se, but simply to demonstrate that molecule per cell counts could be determined from arbitrary units.

- Supplementary Note 3: The calculation focuses on scattering, but what about the spectral response of silica? Could that play a role?

(Due to the changes above, the original Supplementary Note 3 has now been renumbered Supplementary Note 4.)

In this note, we discuss the differences between the light absorbance and scatter properties of cells and microspheres. As light absorbance measurements are a measurement of the absence of transmittance, and therefore depend on both absorption and scatter, it is possible that both play a role. However, we cannot measure these separately. We have assumed that the majority of the effect is due to scatter because it is known that the light absorbance of bacterial cells is largely due to scatter (Walther et al., 1994, we have added this reference into the text). We were also able to fit a model that assumed only a scattering response through the absorbance profile of microspheres (from 200nm to 1000nm) which suggests that the majority contribution is from scatter.

- Supplementary Figure 9: The standard deviations appear to be calculated from just two replicates. With such small numbers, standard deviation estimates are not particularly reliable, and the fact that behavior is not systematic across colors is concerning. There needs to be much more data here, possibly even pulling it to the main text, since quantifying the precision of these assays is critical to establishing the value of this protocol. Likewise, the relationship between buffer and conversion factor should not be gain-dependent, as shown in Supp Figure 10. I suspect that the authors are including the values below the effective sensitivity, and that this is injecting both unnecessary noise and the gain dependence in Supp Figure 10. In the iGEM protocol, these are dealt with by including a blank control, whose variation is used to establish effective sensitivity of the measurement and exclude data below that threshold. I suspect the authors could add a similar control of lysed cells with either no expressed protein or a non-fluorescent expressed protein and use it in the same manner.

The reviewer raises two distinct points here. With respect to the point about assessing the precision of this method, please see the extensive discussion about replicates under point 2. With respect to the point about apparent gain-dependence in the conversion factors calculated using different buffers, see the next few paragraphs.

We regret that we did not explain the context of the gain dependency further in the text. Our methods are based on the methods in Fedorec et al., 2020, whose methods are based on the iGEM methods of Beal et al., 2018. So we have inherited the methods that include the exclusion of values below sensitivity.

We do not observe gain dependence generally. Our point in Supplementary Fig. 10 was to illustrate what happens when proteins are diluted in buffers such as T5N15. We observe that the fluorescence response of the serial dilutions is no longer linear, contrary to what happens in T5N15pi buffer (which includes protease inhibitors). This is of interest since we found that the EC_{max} protein concentration assay can be carried out in either T5N15 or T5N15pi. However, we later found that the fluorescence assay required T5N15pi. We discuss why this might be in the text, on a biological level.

Our point was that in such circumstances, as the fluorescence drops faster than expected, the resultant data points (which are not excluded by the test of at least 1.5x fold difference expectation threshold) result in a bias in the fitting between concentration and fluorescence (Supplementary Fig. 10A) leading to errors in the conversion factors compared to using a reliable buffer (Supplementary Fig. 10B).

The reason this causes a bigger bias at certain gains than others is due to pragmatic aspects of the protocol. We follow the methods of Fedorec et al., 2020, i.e. we prepare a serial dilution that is then subjected to a fluorescence assay at a range of gains (40-120). However, of course, the higher the gain, the more likely that our samples will produce 'overflow' results beyond the range of the plate reader, and therefore a higher proportion of points at these high gains comes from lower concentrations of protein, that produce erroneous data points in certain buffers. Hence the error increases at higher gains, but it does so under these circumstances alone.

- Please provide a mathematical presentation of the key elements data handling in methods or supplemental, not just a link to the R code.

We thank the reviewer for raising this important point. We have now provided this as the new Supplementary Note 6.

Minor points:

- The authors should switch from MEFP to MESF (Molecules of Equivalent Soluble Fluorophore), which is the established nomenclature for the units they are computing. See, for example, <https://doi.org/10.1002/cyto.b.10066>

The reviewer has noted a paper that introduces the MESF (for 'soluble fluorophores') terminology from 2003, and a number of papers have used this notation. However, in synthetic biology circles, the recognised unit would likely be MEFL, for 'fluorescein'. As both fluorescein and FPs are 'soluble fluorophores', we have consciously decided to use MEFP (for 'fluorescent protein') for the purposes of this paper, to distinguish units based on fluorescent proteins from units based on fluorescein.

Our decision is based on a choice for clarity of the message of this paper. Papers that describe analogous methods for fluorescein use MEFL and not MESP notation (e.g. Beal et al., 2018; Fedorec et al., 2020).

- The sentence at lines 75-77 belong in the previous paragraph, on relative units, since calibration to proxy molecules is known to be accurate when the proxy and target are well matched.

Our point in this paragraph is that fluorescein molecules are not necessarily equivalent to GFP molecules. While it is likely there is some approximation, we do not know of a paper that makes this comparison directly – this would require experimental validation. Furthermore, fluorescein is not typically used with thought to actual molecular units, rather, it is more commonly used to standardise units between instruments and laboratories. If used to calibrate green fluorescence of avGFP which is predominantly excited in the UV range, it will likely give results that are far less accurate than for GFPs that are well matched. Therefore, its ability to approximate molecules is not 'absolute', but dependent on the target.

Additionally, most FPs cannot be calibrated by fluorescein, and therefore we believe our point about the inability to extract information about the orders of magnitude of proteins expressed per cell from most plate reader assays stands in general.

- Fit lines should be shown on all graphs where a fit is performed.

We have included fit lines on all plots except those where fit lines are considered detrimental to the visual interpretation of the plot, for example where the data points are of higher importance and multiple fits would lead to overcrowding of the plots e.g. Fig. 3B and Supplementary Fig. 9A.

- Volume conversion of OD is not immune to cellular state, as it will be affected by the degree to which cells are clumped vs. dispersed and by any opaque cellular products.

We use the evidence from Volkmer et al., 2011 to conclude that OD to volume conversion is approximately equal across a wide range of cellular states (across strains, media, and growth rates). We are not aware of any experimental evidence that contradicts this at the moment. While cell clumping is an issue in flow cytometry, that is not the subject of this paper. We discuss the effect of overexpression of absorbing FPs on OD600 estimates in Fig. 6.

- Figure 4B,D: Grey vs. black is harder to differentiate than the colors used in other figures. Please also include a key on the graph, like with the other figures.

We thank the reviewer for pointing this out. We have made the changes as advised:

A calibrate fluorescence per

B

- The iGEM microsphere protocol recommends units of Equivalent Particle Count (EPC) not PEMS. Was there a reason for changing the acronym?

We were not aware of this acronym and don't believe it is widely used. Our microsphere protocols and analytical pipeline is based on the work of Fedorec and colleagues (2020), who use 'particles'. We felt that the use of 'particles' was good, but could do with being more explicitly an 'equivalent' of the MEFL 'molecules of equivalent fluorescein' syntax, hence called the units PEMS (particles of equivalent microspheres).

As with MEFL and MEFP, we believe it is important to be clear about which calibrants are being used for unit conversion, so believe PEMS is clearer than EPC in this respect for this paper. This is especially important while calibrants are evolving and we learn more about their advantages and disadvantages – as Supplementary Fig. 12 shows.

- Figure 5C,D: Do not drop the zero data point. Use a split axis instead.

Data points corresponding to 0% arabinose were not included in Fig. 5C-D as during data processing in the process_plate() function, fluorescence data is normalised to the autofluorescence of negative control cells. This process frequently results in values that approximate zero and are therefore often negative, which cannot be plotted on log plots.

Reviewer #2 (Remarks to the Author):

This paper by Csibra & Stan proposes a novel method for the calibration of fluorescent protein expression, building on prior work in standardising fluorescence measurements so that inter-lab and inter-instrument comparisons are possible. The novelty of the work lies in the use of proteins as the calibrants themselves, rather than fluorescent molecules. This has two main advantages over existing methods: firstly, that it is possible to calibrate red and blue fluorescence in addition to green; secondly that the output values can correspond to a more meaningful measure of protein numbers per cell. This is a compelling argument and has great potential to be a meaningful tool for synthetic biology. I feel that there are a number of improvements in clarity that the paper would benefit from to make its case.

The authors postulate that this has not been attempted previously because of a lack of experience in the required techniques. Arguably the far greater challenge than doing routine protein purification, however, is reliable

quantification which is a pre-requisite for valid calibration. Their results show that the microBCA and A280 assays report values with greater than 2-fold variation in some cases. This seems fairly significantly divergent, and I am unclear on why the A280 result was chosen as the 'reference' value to evaluate the accuracy of the new ECmax assay. It would seem that this variation between assays is large enough to have an impact on calibration efforts, where accuracy is critical (as discussed later on when quantifying protein numbers per cell).

We thank the reviewer for raising this point and agree that accurate protein concentration calculations underly the calibration. We thought long and hard about how to validate protein concentration measurements, as this is rarely done and not conceptually simple. To validate any assay, in principle the results ought to be compared with another assay that is better trusted. After an extensive literature review and a review of commercially available and recommended protein assays, we decided on the A280 assay.

All assays have advantages and limitations, but the A280 assay is one of the most highly trusted and routinely used in protein biochemistry because its limitations are largely down to the fact that proteins need to be highly pure, and need to contain tryptophan and tyrosine residues, which are the ones that absorb in the UV range. In addition, the assay requires the removal of DNA which absorbs highly at 260 nm, and also requires the use of specialised materials for the containers such as quartz, because most plastics absorb in the UV (280nm) range. These are largely practical considerations, and we were able to meet all of them (see Supplementary Note 1). Their advantages are numerous: from only a primary protein sequence, one can directly obtain an extinction coefficient that relates light absorbance at 280 nm directly to protein concentration, which means no internal standard is required. The assay is quick and non-destructive, meaning proteins can be measured for fluorescence or repeatedly measured if desired.

We agree that it was disappointing that the microBCA assay gave results so different from the A280 assay, but we were encouraged by the fact that the A280 and ECmax assays showed high similarity. While the A280 assay does not tolerate the addition of protease inhibitors which contribute to absorbance at 280nm (Supplementary Fig. 5) and therefore make the assay unreliable in its presence, the linearity of the A280 in T5N15 was typically better than for the microBCA. We suspect that the microBCA and A280 assays will show more similarity when higher concentrations of proteins are used, as both these assays had a lower range of sensitivity than the ECmax assay.

It's mentioned that purified FPs are not commercially available (line 87), but various providers do sell purified GFP at least, e.g. Thermo #A42613 and Abcam #Ab84191. Have the authors considered comparing the results from these commercial products to in-house prepared proteins? This may be a necessary quality control if other groups are to use this technique, to verify protein concentration measurements are consistent or even to use directly as the calibrant for those who are only interested in GFP.

We thank the reviewer for this comment. We have spent a significant amount of time considering this point. As reviewer 1 also suggested a similar point, we will respond identically.

Commercially purified FPs.

While we too initially thought that commercially available FPs might be a good way to validate our methods, unfortunately we no longer believe that these invariably have higher reliability or accuracy than purified FPs that we can produce in house with our published protocols. This is due to a number of reasons, including the fact that most GFPs on the market are very different spectrally to 'modern' GFPs and that there is a lack of understanding as well as lack of detailed QC by suppliers selling these proteins. However, as the suggestion to use commercial FPs seems to be a common one, we have set out our reasoning explicitly and in detail in our newly included Supplementary Note 3 (included below).

Despite these points, we identified a few commercially available GFPs that may function reasonably well as calibrants for users who cannot produce their own. We ordered two, one EGFP and one TurboGFP, but unfortunately the EGFP did not arrive in time to test it. The TurboGFP (Pierce Recombinant GFP Protein #88899) was tested in two independent calibrations, described in the new Supplementary Note 3, and presented in the new Supplementary Table 8. From our analysis, we observed that good quality commercial GFPs can provide similar results to in-house produced calibrants.

Paragraph added in the main text:

"The use of commercial fluorescent proteins as calibrants

We were interested in testing whether our purified FPs gave similar RFU to molecule conversion factors as commercially available FPs. Very few FPs are available commercially, and the majority are green FPs, so we focussed on those. We compiled a table of available GFPs to ascertain the best candidates to order (Supplementary Table 7). Surprisingly, many were based on first generation GFPs that preferentially excite in the UV range (unlike GFPmut3 or sfGFP), had incomplete datasheets lacking protein sequence, brightness or

spectral information, or were not subject to explicit quality controls (Supplementary Note 3). This suggested their production may be less rigorous than the methods described in this paper and makes them difficult to recommend as calibrants (Supplementary Note 3). Nonetheless, using a commercial TurboGFP, we obtained a relative conversion factor of 95.0% compared to the value obtained using our purified mGFPmut3, which suggests that FPs from in-house and (carefully selected) commercial sources may be used interchangeably (Supplementary Table 8)."

Text added in the supplementary material:

"Supplementary Note 3. The use of commercial fluorescent proteins as calibrants

As described in the main text, we were interested in testing whether our purified FPs gave similar conversion factors as commercially available FPs in order to validate our methods and results, with a focus on GFPs. The result of our efforts to compare commercially available GFPs is provided in Supplementary Table 7.

Unfortunately, we observed several problems with the use of these products as reliable calibrants. First, the majority of the available FPs are either the original "wild-type" *Aequorea victoria* GFP (FPbase: avGFP/1XF1B) – which is almost never used in modern cell biology – avGFP-related but obscure proteins such as alphaGFP (FPbase: B28N7) or Q80R (FPbase 799YV), or commercially-developed proteins such as 'GFPSpark' (not on FPbase). Some of these proteins do not have FPbase entries or recorded fluorescence spectra, and others, such as avGFP, have significantly different spectra from the 'modern' GFPs like GFPmut3 or sfGFP, absorbing light maximally in the UV range, rather than the blue (488 nm) range. We also found that some FPs were poorly annotated in a way that suggested a lack of care in their datasheets (see Supplementary Table 7), and technical support staff occasionally described avGFP as 'normal' or 'standard' GFP – suggesting an erroneous perception that such GFPs are widely used, or that the identity of the GFP in question is not considered important for most applications. Some suppliers confirmed they do not test their FP batches for fluorescence or carry out QC on their fluorescence activity.

These observations suggested that we would need to validate commercial FPs against our FPs, rather than the other way round. We selected the TurboGFP available from Pierce for testing, and performed calibrations in T5N15pi buffer with the ECmax assay (Supplementary Table 8). Our results confirm that this FP produces similar results to our mGFPmut3 calibrants, with a mean conversion factor of 95.0 % compared to the conversion factor of mGFPmut3. This suggests that users can use commercial FPs to gain approximate conversion factor estimates provided they check for a good match between the properties of their FPs and those available.

Of interest is the fact that in the few weeks since we ordered the Pierce TurboGFP, this product has already been discontinued. For this and the above reasons, we would generally recommend that users make their own calibrants. These are easy to produce, bespoke to the specific FP used in users' own cells and circuits and can be affordably produced in high quantities. Moreover, if users purify FPs and run an SDS-PAGE to ascertain purity, they will have done as much QC as is typically done on commercially purified FPs. As our protocol involves taking an entire absorbance spectrum, this can act as an added QC step to verify FP quality against expected spectra on FPbase. Dilution in protease-inhibitor free buffer and the use of UVclear plates allows users to also check the protein's concentration using two different methods, providing yet another QC step over a commercial protein."

Our new Supplementary Table 8 illustrates the relative conversion factors obtained from the mean of two independent repeats of a calibration experiment using TurboGFP (the stock tube contained only enough for two repeats), compared to the mean conversion factor obtained from in-house produced purified mGFPmut3.

The new Supplementary Tables 7 and 8 are provided as attached files.

Supplementary Table 7: Comparison of commercially available GFPs. The properties of purified GFPs available as of May 2022 listed in order of suitability as calibrants for calibration of the mGFPmut3 protein used in this work. The left of the table compares the GFPs in terms of their identity with FPbase-recorded protein sequences and fluorescence properties, and the right in terms of the purification and QC procedures of the supplied proteins.

Supplementary Table 8: Relative conversion factors. Conversion factors (relative fluorescence units/molecule number) from purified, sonicated lysate and chemical lysate calibrants are all shown as relative to the mean of the purified equivalent fluorescent proteins (purified mCherry values were compared to the mean of all purified mCherry values, and sonicated mCherry lysate values were also compared to the mean of purified mCherry values, etc). One anomalous value (mTagBFP2 purification set1) was removed from the calculation of the mean as described in the text. TurboGFP is shown as compared to the purified mGFPmut3 samples. Sets here refer to independent replicates. Abbreviations: stdev, standard deviation; CV, coefficient of variation.

The description of the method development is a little unbalanced and could be more instructive in guiding the reader on how to perform this calibration themselves. There is a detailed description of the optimisation of two standard protein quantification techniques (microBCA and A280), after which the authors choose to use a third assay (ECmax). This makes it confusing to understand which is really necessary. Since they have discovered that the simpler method of using cell lysate and measuring ECmax provides equally good results, I feel that it should be more explicit that this is the recommended protocol rather than providing all three in the methods (line 563-573). Describing it as a 'hack' implies performing the full-length protocol with three quantification assays is still preferable, even though it doesn't appear from their results that there is any increase in accuracy for doing so. Perhaps the 'best' protocol could be included in the supplementary for completeness and the others can be linked to on [protocols.io](https://www.protocols.io) for reference.

We thank the reviewer for this comment. Investigations into the earlier protein assays were a requirement for concluding that the ECmax method performs best, by allowing us to make comparisons between the different methods. The description of it as a 'hack' is merely meant to signify that it is the quickest and easiest of the methods tested, and we only use this word twice. In both cases the relevant sentences emphasise that this method is the best method in terms of *both* simplicity and performance. Nonetheless we have rephrased the two sentences.

In the introduction, we originally wrote: "We also present a 'hack', which acts both to simplify the method to remove the requirement for protein purification, and to make it more sensitive and robust." This has been replaced with "We also present a novel fluorescent protein assay, which acts both to simplify the method to remove the requirement for protein purification, and to make it more sensitive and robust." And in the discussion, we wrote: "We also discovered a methodological 'hack' to obtaining FP concentrations using the extinction coefficients at their maximum excitation wavelength, the ECmax assay", which was replaced with "We also discovered a novel methodological shortcut to obtaining FP concentrations using the extinction coefficients at their maximum excitation wavelength, the ECmax assay".

We have followed the reviewer's suggestion for clarifying the Methods section which now reads:

"The FPCount protocol consists of calibrant preparation followed by a protein assay (for protein concentration) and a fluorescence assay (for protein activity). Each step is described in the following sections below. In addition, we have provided step-by-step instructions for the recommended FPCount protocol in Supplementary Note 5. This protocol consists of the preparation of calibrants as fluorescent proteins in cell lysates prepared by sonication, and the use of the ECmax assay as the protein assay. This protocol is also available on [protocols.io](https://www.protocols.io), at <https://www.protocols.io/view/fpcount-protocol-in-lysate-purification-free-protobzudp6s6> (55). We also detail two other calibration protocols on [protocols.io](https://www.protocols.io): the full protocol conducted for Fig. 3, using FP purification and three protein assays for cross-validation, available at <https://www.protocols.io/view/fpcount-protocol-full-protocol-bztsp6ne> (53), and a shorter protocol requiring purification but using the ECmax assay only, available at <https://www.protocols.io/view/fpcount-protocol-short-protocol-bzt6p6re> (54). "

As stated above, we have copied the recommended protocol into a new Supplementary Note 5.

If someone did wish to cross-validate their results by performing the longest protocol, what is the cut-off for acceptable outcomes (e.g. 2-fold variation between assays, or 3-fold, etc.)?

This is an interesting question. Given our results and our response to the reviewer's first point above, we suspect that the use of relatively low quantities of protein in the dilution (i.e. with highest concentration between 10-30ng/ μ l) will likely result in ~2-fold differences between microBCA and the other assays. We suspect that starting from lower concentrations will result in larger discrepancies, since microBCA is less sensitive than the ECmax assay, but the use of higher concentrations should lead to higher degrees of similarity. Rather than suggest a specific cut off, since we have not investigated this point, we suggest instead that it is better to be aware that the use of different protocols may have an effect in the order of 2-3-fold depending on buffer conditions, assays etc.

The required accuracy depends on the application and is therefore up to the user. If a general idea of abundance is all that is required (e.g. a ballpark estimate to determine if a protein is at low (<100) medium (100-1000), high (10,000), or very high abundance like in figure 6 (100,000+)), then those experiments arguably require far lower accuracy than those in which a ~2-fold difference is required to be distinguishable.

The second half of this paper looks at the relevance of this work for metrology in the field of synthetic biology. The main strength of the existing fluorescein calibration method is that it has been validated across a large number of different users with different instruments (ref 15), and until this has also been demonstrated for protein calibrants it should be mentioned as a limitation. In order to facilitate that type of inter-lab validation, it would be helpful to make these FP SEVA plasmids available on a platform such as Addgene, so that people can reproduce

this protocol and compare their MEFP/PEMS measurements to the published results.

We thank the reviewer for this point, and have added a point in our discussion about how the FP method compares with the fluorescein method, and about the fact that inter-lab studies using this method would be valuable:

“Current users of fluorescein calibrations may be interested in how this method compares with calibration using a serial dilution of fluorescein. The protocol for fluorescein calibration is certainly cheaper (£0.23 per calibration using details from Beal et al., 2018) owing to the fact that fluorescein is a low-cost fluorophore, and simpler, as the use of commercial calibrants obviates the need for calibrant preparation or concentration determination. The key advantage of FP calibration lies in its ability to be used to directly convert arbitrary units directly to molecules of one's desired FP, coupled with the freedom to produce bespoke calibrants from any desired FP, or indeed multiple FPs, from across the spectrum, whether or not they match the properties of fluorescein. While commercial FPs cost an average of £39.05 per calibration (assuming use of 10 µg FP per calibration, Supplementary Table 7), the estimated cost of producing one batch of FP calibrants in lysate is only £10.18, from which the standard yields are approximately 100 µg FP (using only a fraction of the culture). While FPs can display condition-dependent fluorescence, as discussed above, these effects can be minimised: indeed, they can be used to test the temperature or pH sensitivity of a user's FP under controlled conditions. Furthermore, fluorescein and other small molecules also display condition-specific fluorescence, though this point is rarely considered (Le Guern et al., 2020). While it is difficult to directly compare the precision of this method with published data on fluorescein, the FPCountR protocol is clearly highly reproducible. Calculated conversion factors are typically within $\pm 10\%$ of the expected value and coefficients of variation between independent replicates are typically below 0.1 (Supplementary Table 8). Following on from the exemplary work on fluorescein (Beal et al., 2018), there is clear potential for future inter-lab studies to extend our understanding about the accessibility and reproducibility of the FPCountR protocol.”

In addition, we have arranged for our plasmids to be made available via Addgene and have added the relevant information to the methods section:

“All three pS381 plasmids used in purifications are available from Addgene under the IDs 186733-186735.”

I appreciate that fluorescence quenching is addressed as a potential area of concern. However, the method chosen to measure this, mixing purified FPs with cell culture, does not seem to account for the effect of fluorescence being altered by FPs being inside the cell. Ref 25 looks at this so-called "inner filter effect". Can the authors comment on whether they would expect a difference in observable fluorescence between FPs in solution with cells, and FPs expressed within a cell?

Yes, the phenomenon we measured was effectively the “inner filter effect” described in ref. 25. We decided on an experimental methodology that added cells to purified FPs since the major mode of signal attenuation postulated by previous authors on this subject is essentially light scatter due to cells. Light scatter would reduce the intensity of light at the excitation wavelength hitting an FP and will also deflect emitted fluorescence from hitting the detector. We expect the magnitude of this effect to be the same inside cells as outside cells, and to depend only on the concentration of cells in the well.

Other authors have described other methods of doing this, which we initially considered. Experiments measuring the fluorescence of FPs in cells pre vs. post lysis (e.g. Hirst et al., 2015) are technically challenging due to their requirement to get 100% cells lysed. The authors suggest the use of detergents rather than sonication for this purpose; however, the addition of detergents presents a large challenge for accurate quantification since they can affect the absorbance of the resultant lysate which can be detrimental to the accuracy of FP quantification (see for example Fig. 3F). Detergents are also very difficult to remove completely. Therefore, on balance, we decided on the current assay as the best way of collecting reliable data on the inner filter effect / quenching effect.

The efforts to quantify the impact of red (and apparently also blue) fluorescent proteins on OD700 measurements are interesting and should be very informative for others in the field.

I recommend this paper for publication as it proposes a useful new tool, and the authors argue persuasively that the best and most relevant calibrant for fluorescence measurements would be the proteins themselves. I think it is better conceived primarily as a methods paper, and as such there should be an effort made to communicate more clearly how this method can be performed and validated by others.

Further minor comments:

- FPbase is frequently referenced but readers may not be familiar with this resource. A short description of this database and the reliability of the data would be helpful when it is first mentioned.

We thank the reviewer for pointing this out. We have added the following:

“FPbase (www.fpbases.org/) is an open-source, community-editable database of fluorescent proteins and their properties. Each FP in the database contains its own page with a structured set of properties, such as its primary protein sequence, extinction coefficient and fluorescence spectra. Essentially all commonly used FPs have entries on this database, along with rarely used variants, and these are accessible via its API.”

- The caption for figure 2A includes information that is exclusively mentioned there, e.g. the choice of His-tag as affinity tag and expression conditions. I feel this is better suited to be moved to the main text.

We have moved these details back to the main text, and slimmed down the figure legend for Fig. 2A accordingly. The relevant text now reads:

“The use of high-copy vectors and overnight expression was designed to maximise protein production, and the temperature was dropped to 30 °C to minimise misfolding. Cells were lysed using sonication to avoid the requirement to add chemical components that may interfere with downstream processes, such as EDTA (with His-tag purification), detergents (with protein quantification), or unknown components of commercial lysis reagents. Insoluble proteins were removed via centrifugation and SDS-PAGE was used to confirm that the majority of the expressed FP was in the soluble fraction (Fig. 2C). Proteins were purified using His-tag affinity purification, as His tags are small in size, making them unlikely to compromise fusion protein function. Cobalt resin was used as the affinity matrix as it has higher specificity for His tags than nickel resin, and was therefore expected to co-isolate fewer impurities. The quality of purified FP calibrants was verified by SDS-PAGE and fluorescence excitation and emission scanning (Fig. 2D-E, Supplementary Fig. 1).”

- Fig 2E shows 2 peaks corresponding to the excitation and emission spectra. It would be helpful to label them as such.

We thank the reviewer for pointing this out and have added labels to this figure.

- Fig 5C-D Can the authors suggest an explanation for why FP concentration decreases over time in the uninduced condition?

The observed ‘reduction over time’ in the uninduced condition was likely due to transcriptional leakage of the high copy vector during prolonged stationary phase growth in overnight cultures. We have repeated the experiments in Fig. 5C-D to study if this effect is reproducible and have found that the leakage is not reproducible. At the same time, we observed lower levels of mCherry per cell in the repeated experiments.

Rebuttal Fig. 4. Comparison of independent repeats of arabinose titrations with pS381_ara_mCherry. mCherry expression from a high (pS381, colE1) copy vector was induced at a range of arabinose concentrations and quantified in a timecourse assay as molecules per cell in a calibrated plate reader. Data from each of three independent experiments (1,2,3) is presented side by side. Data was collected from three biological replicates, each of which is plotted.

We suspect therefore, that the discrepancies between the mCherry molecules per cell across the repeats is largely due to the fact that in the first experiment we observed a high level of leaky expression in our overnight cultures (in the order of 1000s molecules per cell at time point 0, when arabinose was added), that inflated the total mCherry values throughout the time course in experiment 1. The other experiments show lower overall mCherry values at all timepoints due to mild to undetectable levels of leakage (mCherry levels at 0% arabinose (in red) are often so low as to not be plottable even on an extended y axis starting at 1 molecule/cell).

We therefore decided to replot the data in Fig. 5C-D using data from mCherry experiment #2 for the top panel, and averages from experiments 2 and 3 in the bottom panels, as the newer data seem to be more representative of the behaviour of these plasmids. In addition, as our calculated average conversion factors for each FP have been updated with new data since our original manuscript, particularly for mTagBFP2, we have reprocessed the data with the new conversion factors to take into account these changes. The resultant new Fig. 5CD is as follows:

Figure 5. Absolute quantification of *E. coli* timeseries data in molecules per cell.

C-D. Absolute protein quantification in molecules per cell (C) and molar concentration (D). mCherry expression (*top*) from medium (pS361, p15A) and high (pS381, colE1) copy vectors was induced at a range of arabinose concentrations and quantified in a timecourse assay in a calibrated plate reader. Data was processed as described in (A) and cell estimates based on microsphere calibrations were used to calculate per cell values (C), or OD-specific cell volume data from Volkmer et al., 2011 was used to calculate molar concentrations (D). Data was collected from three biological replicates, each of which is plotted. (*Bottom*) mCherry expression from the top panel is compared with mTagBFP2 expression from an identical assay, plotted against arabinose concentration at 420 min post induction. Displayed points show the mean values from two independent experiments, each of which tested three biological replicates. Error bars indicate standard deviations.

In order to bring the text in line with the new data, we have edited the following sentences:

“Using these vectors, we obtain figures for mCherry abundance that vary between about 900 to 70,000 molecules per cell for p15A, and 200 to 200,000 for colE1.”

“Interestingly, measurements from identical SEVA vectors revealed that while the FP abundances were in the same order of magnitude, mTagBFP2 accumulated to higher levels per cell than mCherry by 3.8-fold on average, despite sharing the same promoter, 5' untranslated region, ribosome binding site and N-terminal protein sequences (Fig. 5C-D).”

We have also removed the sentence: “Leaky expression could be quantified in the colE1 vector as contributing hundreds of mCherry molecules per cell even without induction, whereas in the p15A vector it was merely ten copies per cell.”

- For fluorescein calibrations it is recommended to repeat the process and calculate new conversion values regularly to account for instrument drift. Presumably this is also true with FP calibrants. If so, is it possible to use the same purified protein after a period of months, or would they be expected to degrade in storage over time?

We thank the reviewer for raising this interesting point. We have heard conflicting opinions on the likelihood of instrument drift for plate readers, with some recommendations suggesting that minimal drift is expected over even a year of use.

Regarding the stability of the calibrants, we have not been able to do extensive testing on this yet, as long-term studies are out of scope for this manuscript. However, we have now tested two batches of calibrants in lysate to look at whether the conversion factors obtained after storage at 4°C change over time. The results suggest reasonable stability for mCherry over a 6-week period, and excellent stability for mGFPm3 and mTagBFP2. We have included this data as the new Supplementary Fig. 11C.

Supplementary Figure 11. Performance of FPs in lysate as calibrants. C. Stability of two sets of sonicated lysates over a period of storage at 4°C. Two of the batches of calibrants prepared for A-B were stored at 4°C and remeasured over a period of six weeks. Plots show their measured conversion factors relative to the conversion factors measured at 0 weeks.

We have included the following in the text:

“Repeated testing of calibrants prepared by sonication suggests that FPs maintain stability in lysates when stored at 4 °C for a number of weeks (Supplementary Fig. 11C).”

- Supplementary figure 1B is missing labels for the FPs.

We thank the reviewer for spotting this. The missing labels have now been replaced.

Reviewer #1 (Remarks to the Author):

I thank the authors for providing additional details and for producing a larger number of replications of protocol execution in order to better determine precision. The evaluation of commercial fluorescent proteins was particularly surprising and interesting (if anecdotal), and systematic study of that issue might be an interesting paper on its own.

The core concerns of my prior review, however, remain unaddressed, to wit:

- Can the authors provide an experimental *accuracy* determination? In order to determine accuracy, one generally needs to compare the method under development to at least one other independent method for performing the measurement. This still has not been done. The authors have produced estimates of fluorescent protein count, and have produced estimates of the precision of their method, but the manuscript still does not contain anything that allows us to determine how close those estimates are likely to be to the true value.
- The authors have still not provided any means of determining whether the protocol has been executed correctly. While I am quite willing to believe that the authors have executed correctly and with great care, the same will not be true for new users of the method whose understanding will be imperfect. This issue is particularly important because of the complexity of the method and the ample opportunities for opaque failures during its execution. What process controls can be used to determine whether a user of the method should trust the results that they have generated?
- The authors continue to claim that their work should not be compared with dye-based methods, because dye molecules have a different spectrum. This overlooks the obvious option of computing a conversion from the excitation and emission spectra of the two molecules and the excitation and emission conditions of measurement. While the spectral conversion method is imperfect, so is FPCountR, and some of the variation that is now exposed in the revision makes me guess that the accuracy of FPCountR is likely on the order of 2-fold to 4-fold off from the true value, which I would guess may be less accurate than the spectral conversion method.

Due to these shortcomings, at present I would advise against anyone making use of FPCountR, because:

- 1) I do not know whether its protein counts are close to correct or an order of magnitude high or low, and
- 2) there are no process controls that can help a person determine whether they have executed FPCountR correctly or have bungled it.

I do think there is a good chance that FPCountR is indeed a significant advance, and if this can be established, I would love to make use of it. Determining accuracy and providing process controls, however, are sine qua non for a calibration method.

There are some other issues that I believe still need to be addressed as well, but they are all much smaller and could be handled through minor revisions if the core concerns can be addressed.

Reviewer #2 (Remarks to the Author):

The authors have thoroughly responded to my feedback and I feel they have addressed the major comments in my initial review. My only minor comment is that the labels `em_max` and `ex_max` in Supplementary Table 7 are the wrong way around. I am happy to recommend this paper for publication.

REVIEWER COMMENTS

Reviewer #1 (Remarks to the Author):

I thank the authors for providing additional details and for producing a larger number of replications of protocol execution in order to better determine precision. The evaluation of commercial fluorescent proteins was particularly surprising and interesting (if anecdotal), and systematic study of that issue might be an interesting paper on its own.

We thank the reviewer for their detailed and thoughtful comments in the first round and agree that the evaluation of commercial fluorescent proteins could be studied further.

The core concerns of my prior review, however, remain unaddressed, to wit:

- Can the authors provide an experimental *accuracy* determination? In order to determine accuracy, one generally needs to compare the method under development to at least one other independent method for performing the measurement. This still has not been done. The authors have produced estimates of fluorescent protein count, and have produced estimates of the precision of their method, but the manuscript still **does not contain anything** that allows us to determine how close those estimates are likely to be to the true value.

We thank the reviewer for this excellent point about accuracy, and apologise if we haven't been clear enough that we both agree that accuracy is an important consideration for a calibration method, and that we have indeed considered it in a great deal of detail.

The question of the accuracy of FPCountR is addressed in 7 places in the paper. To respond to the reviewer's concern that we have not examined accuracy at all, we will consider each element in turn in our response, with reference to the places in the paper where this has already been considered and/or tested. In addition, we describe 2 extra accuracy tests and 1 extra accuracy estimate from the literature.

1. Accuracy of FP quantification of calibrants

Accuracy of FP quantification of calibrants – general considerations

We thought long and hard about how to ascertain what the 'true' values of protein concentration were for each batch of FPs prepared as calibrants, in order to estimate the accuracy of our methods. Ultimately, this question is a bit of a chicken and egg problem, since the commonly accepted answer to the question 'how can I maximise the accuracy of my protein assay?' is to measure your protein of interest in parallel with a set of suitable standards that should ideally be purified versions of your protein of interest, purchased from a reliable supplier. As FPCountR aims to establish a method for making such standards (rather than quantifying a certain sample by buying in the standard), and as commercial sources do not appear to follow more stringent quality control than we have done, we are left in the rather curious position of requiring purified GFP standards of higher quality than ours (that may not commercially exist) to verify our own purified GFP standards. If it is likely that there is no better-quality preparation than your own, a comparison with other standards is not necessarily a good validation method.

Accuracy of FP quantification of calibrants – using different methods

Pragmatically therefore, instead of comparing our GFP preparations with external standards, we initially opted to instead use two trusted protein concentration methods and to compare them to each other (Fig. 3A-C). In other words, we explicitly set out to validate the quantification method with an independent method, exactly as the reviewer recommends. (To directly quote the text: "We sought to verify the accuracy of the BCA assay by re-quantifying our FPs with a second method that is likely to give reliable concentration estimates...")

There are numerous assays for determining protein concentration, and we carefully considered which one to use and how to ascertain what the 'true' values of FP concentration were, given each assay's inherent limitations. In the end we opted for the A280 assay and the BCA assay, because the available evidence in both the academic literature as well as from technical pages on protein quantification guides from reagent suppliers

suggested these might be the two most reliable assays for fluorescent proteins. The A280 assay is the most widely used assay in protein quantification, commonly accepted to produce results of high accuracy as long as the proteins have multiple residues that absorb light at 280 nm (mCherry and mTagBFP2 have 15 such residues, while mGFPmut3 has 14). It does not require its own standard, as extinction coefficients depend only on the primary sequence of the protein, and many of the standards sold with other assay kits are themselves quantified using the A280 assay (Thermo Fisher, personal communication), suggesting it is the most trusted method for protein concentration quantification. The BCA assay is also widely used, and while it does require an internal standard, it has reportedly low protein-to-protein variability in its accuracy (as opposed to the Bradford assay: Noble et al., 2007).

As described in our results section: “We sought to verify the accuracy of the BCA assay by re-quantifying our FPs with a second method that is likely to give reliable concentration estimates. While a wide variety of protein assays exist, the only widely-used ‘absolute’ assay that does not require a calibrant is the A280 assay. ... We sought to conduct a systematic assessment of the BCA and A280 methods by testing three spectrally distinct FPs in two buffers, assessed with both assays in parallel (Fig. 3). ... The results of this comparative test are shown in Fig. 3B-C, Supplementary Fig. 9 and Supplementary Tables 1-3. Broadly, the results from each assay validate those of the other assay: the measured concentration of each FP using the microBCA and A280 assays are within 2-fold of each other for most samples (Fig. 3C) and apparent linear ranges reach 1 ng/μl for most dilution series (Fig. 3B).”

Thus, the initial results suggested the protein concentration estimates from either assay are at most about 2-fold out. This may have been our conclusion had we stopped there. However, as we found in parallel that the third assay, the ECmax assay, produced very similar results to the A280 assay (typically predicting concentrations matching the results from the A280 assay to 80-100%) in T5N15 buffer that doesn’t include 280nm-absorbing additives (see Supplementary Table 1), we suspected that the A280 and ECmax assays were, on the balance of probabilities, likely to be more accurate, and the microBCA was likely to be less accurate. This gives us an accuracy estimate for the ECmax assay of <20% compared to the reference value from the A280 assay.

Another strand of evidence that the A280 assay was likely to be more accurate at the low protein concentrations we were working with (1-10ng/μl range), was that the linear regressions between the known dilution and measured concentration generally fit better to the data points (Supplementary Fig. 9A). We have now included the root mean square error (RMSE) statistics for each of the fits for every protein quantification assay tested in Supplementary Fig. 9A to underline this conclusion (Supplementary Table 2). These show that the A280 assay produces better fits (lower RMSE scores) than the microBCA assay, suggesting it may be more reliable at the relatively low concentrations used in these assays.

Method	Buffer	RMSE ¹
microBCA	T5N15	3.40351984
microBCA	T5N15_pi ²	3.69610429
A280	T5N15	0.62756108
A280	T5N15_pi	3.14890343
ECmax	T5N15	0.12288337
ECmax	T5N15_pi	0.16235333

Supplementary Table 2: Root mean square error statistics for protein quantification assay data. ¹Mean RMSE across all proteins and data sets in Supplementary Figure 9A. ²²The ‘pi’ notation indicates the inclusion of protease inhibitors.

We have clarified the appropriate paragraphs in the text as follows: “Overall, the A280 assay produces data that fits better to a linear regression than the microBCA assay, suggesting it may be more reliable at the relatively low concentrations used in these assays (Supplementary Table 2).” “In addition, predictions from the ECmax assay closely match those from the A280, typically predicting concentrations matching at 80-100 %

those of the expected result (rather than 170-220% for microBCA; Fig. 3C, Supplementary Table 1), suggesting an error rate of <20% compared to the reference value from the A280 assay.”

Accuracy of FP quantification of calibrants – using external calibrants

1) TurboGFP

Since the original work, and to respond to the first round of reviews, we obtained a commercially purified TurboGFP and used it to calibrate our instruments using our method. As discussed previously, this protein produced conversion factors from RFU to protein that were 95% of the values obtained with our mGFPmut3 (i.e. within 5% of our result) (Supplementary Table 5). While we suspect our FPs were of higher quality, this result supports our claim that our FP quantification methods (used for mGFPmut3) are reasonably accurate, since the independent methods for FP quantification used by at least one biotech company (for a similar GFP to mGFPmut3) produced results within 5% of the results we get with in-house produced FPs and in-house developed methods.

2) Fluorescein

We have now completed a comparison of mGFPmut3 calibrations with fluorescein in addition to TurboGFP. Four independent calibrations were conducted, and the relative conversion factors compared to mGFPmut3 were calculated. Without applying a correction method, the relative conversion factors from fluorescein were 77% higher than those measured for mGFPmut3 (Supplementary Table 5). After a correction that took into account spectral differences in the relevant channels, conversion factors from fluorescein measured 33% lower than those measured for mGFPmut3 (Supplementary Table 5).

(For interest, we also applied the spectral conversion method to TurboGFP, obtaining values that suggest an error of 10 % (see updated Supplementary Table 5).)

Supplementary Figure 12. Comparison of the conversion factors of mGFPm3 and fluorescein. A. Fluorescence spectra. Excitation (ex, dashed lines) and emission (em, solid lines) spectra of mGFPm3 (red) and fluorescein (black). The excitation (blue-) and emission (green-) shaded areas represent the bandpass filter wavelengths of our GFP-specific filter sets (respectively, 480/20 and 535/25 nm). **B. Relative conversion factors.** A serial dilution of known concentrations of fluorescein was prepared and fluorescence intensity values were taken with the same instrument, filter set and gain as the mGFPm3 calibrations done previously. Calculated conversion factors were adjusted for the expected difference in brightness and excitation/emission in the ‘GFP’ filters. Each point represents one independent experiment taken from a serial dilution prepared in duplicate.

Thus, if normalised for spectral differences, fluorescein calibration gives values very similar to mGFPmut3. Such a cross-validation of a more direct calibrant (the same FP) with a more established calibrant (fluorescein) has two potential implications. First, if we work on the assumption that calibration protocols with fluorescein are more likely to be accurate than those with FPs, based on the idea that methodological simplicity is the most important factor for accuracy, this suggests an error for the FPCountR method of about 33% (up slightly from the 20% estimated from the A280:BCA comparison). Alternatively, if we work on the assumption that accuracy is best obtained by using identical molecules as calibrants, this experiment provides the first experimental demonstration (to our knowledge) that using fluorescein as a calibrant could have the potential

to convert RFU to units of protein. Further work may be necessary to make a general conclusion, given the differences in properties between small molecule fluorophores and FPs (as considered in the Discussion).

Along with the above figure, we have added the following to the Result section:

“FPCountR produces comparable conversion factors to commercially available calibrants

... Further, calibrations using the small molecule fluorescein produced conversion factors with only 33% error compared to mGFPmut3, provided that the spectral differences between mGFPmut3 and fluorescein were accounted for (Supplementary Fig. 12, Supplementary Table 5), showing decent comparability between a protein and small molecule calibrant. To our knowledge, this is also the first experimental validation that fluorescein may under certain conditions allow the conversion of RFU not just to molecules of fluorescein but also to approximate molecules of protein.”

2. Accuracy of cell number quantification

Accuracy of cell number quantification – using different methods

We considered this both in the original text and in our response to the first round of reviews, namely, that there is approximate equivalence between ‘per cell’ counts in a plate reader vs a flow cytometer when cell counts are calibrated using microspheres. We did not carry out these validations because they have been completed by at least two groups previously.

As we wrote in the original text: “As to microsphere calibration, cross-comparison with flow cytometry data suggests cell count estimates from microsphere calibrations are reasonable (Fedorec et al., 2020; Beal et al., 2020)”. Using data from these publications, we have now estimated the cell count accuracy from microsphere calibration by comparing the MEFL/particle values the authors obtained from flow cytometry vs. plate reader data on identical samples.

Plasmid	Reference	Flow cytometry (MEFL/particle)	Plate Reader (MEFL/particle)	Accuracy
J23101	Fedorec et al., 2020	443395.03	597775.74	0.74
J23106	Fedorec et al., 2020	37729.93	48437.81	0.78
J23117	Fedorec et al., 2020	301.51	600.25	0.50
J23101	Beal et al., 2020	254432.60	165887.72	1.53
J23106	Beal et al., 2020	98936.38	93785.61	1.05
J23117	Beal et al., 2020	978.84	3062.44	0.32

Rebuttal Table 1. Cell count accuracy with microsphere calibration, assuming flow cytometry values are accurate. Data obtained from Fig. 3 in Fedorec et al., 2020 (courtesy of Alex Fedorec).

Cell count accuracy was calculated as MEFL/particle (by flow) divided by MEFL/particle (by plate reader). Assuming the fluorescein calibrations on both instruments were accurate, and the error in the MEFL/particle measurements is the result of cell count discrepancies only, allows the elimination of the MEFL terms and results in “cell count accuracy = Particles (by plate)/Particles (by flow)”. Expressed this way, and assuming that flow cytometer particle counts are accurate, it seems that plate reader cell counts as quantified with microsphere calibration are reasonably accurate. While the measured accuracy values vary, the mean accuracy in the Beal 2020 study was 0.97, while that from the Fedorec 2020 study was 0.67. As our data was collected using identical microspheres and instruments as the Fedorec study, our current best estimate of microsphere calibration accuracy suggests it may result in about 33% error.

3. Overall accuracy of protein molecules per cell

We considered the overall accuracy question in detail in our Discussion (paragraphs 6-9). In brief:

- We noted that our estimates fall in the correct orders of magnitude expected for protein counts per cell from proteomics experiments (paragraph 6)

- We considered factors that limit accuracy when absorbance and fluorescence measurements interfere with each other, and described how our methods allow us to increase accuracy here (not quantitatively considered before in such detail) (paragraph 7)
- Finally, we concluded that fluorescent protein quantification was likely to be highly accurate, but noted that cell counts might limit the accuracy of the estimates in certain cases (paragraph 9). We concluded that, particularly when expressed as molar concentrations, the obtained figures should be approximately accurate (within 2-fold) under the assumption that the OD-specific cell volume doesn't vary significantly between samples or over time.

To this, we have now added an overall estimation of protein per cell count accuracy:

- “The overall error in the accuracy of protein per cell quantifications using FP- and microsphere-calibrated microplate readers can be estimated as the sum of its components: 20% (from protein quantification error, Fig. 3C, Supplementary Tables 1-2) and 33% (from cell count error deduced from flow cytometer counts, Fedorec et al., 2020), which results in approximately 50% error. This should be accurate enough to allow protein per cell quantification not just to the correct order of magnitude (ie. to estimate whether protein abundance is in the 100s/cell or 1000s/cell), but also suggests that quantifications are likely to be accurate to within two-fold of the true values. This should be more than adequate for most applications, including estimations of the relative magnitude of two or more FPs, as well as for genetic circuit modelling. Additional errors, such as those from cell quenching or from the comparison of OD600- vs OD700-quantified cell counts, can be avoided by the use of the FPCountR analytical R package. Others, such as those from pH-sensitive FPs, can be avoided by using pH-insensitive FPs.”

Overall accuracy of protein molecules per cell – further considerations

The reviewer is technically correct to say that we have not directly validated our *overall* ‘molecule per cell’ count estimates with an independent method. The reason for this is that we don’t believe that a reliably more-accurate method exists that would be suitable for such a validation*. This is why we have focussed on the validation of the component methods separately, namely (a) FP quantification, (b) cell count quantification, and (c) their interactions. The obtained counts are correctly viewed as *estimates*, but that is arguably the case with any biological measurement.

**Mass spectroscopy might be the most obvious candidate for obtaining independent estimates of absolute protein molecule counts per cell. However, mass spectrometry too depends on purified calibrants for obtaining absolute numbers, in other words, again we would be relying on the rather circular logic of using our FP preparations to calibrate both a microplate-based method and an independent method: the results of such a comparison would again not have the power to determine the accuracy of the quantification of our FP calibrants. We have also received direct advice from experienced LC-MS (liquid chromatography–mass spectrometry) practitioners that UV spectroscopy (i.e. the A280 assay) and fluorescence methods are considered more reliable ways of getting absolute concentration measurements of GFP than LC-MS. They estimated that the set up and optimisation of LC-MS for this purpose would be expected to take months and be a project in its own right.*

Addendum: Overall accuracy of protein molecules per cell – validation of results from Fig. 5

To follow up on the reviewer’s apparent concern that our experimental results from Fig. 5 (where we looked at molecule numbers from FPs expressed from identical vectors and found a 3.8-fold difference between mCherry and mTagBFP2 in the p15A vector) reflect a quantification error rather than an observation of a true difference in abundance due to the different FPs, we offer a few supporting lines of evidence. First, in our experience it is often the case, when coding sequences for one protein get replaced with coding differences for a second gene, that resultant abundances of the two proteins vary by a large factor. Second, when we use these same constructs for calibrant production, they reliably result in different yields of protein. This is most apparent by SDS-PAGE which was used to check the quality of our purifications:

B

FP	FP density	Total protein density	FP fraction (%)
mCherry	15,004	209,654	7.16
mGFPmut3	38,528	198,145	19.44
mTagBFP2	31,696	243,371	13.02

Rebuttal Fig. 1. Fraction of cellular proteins that is fluorescent protein. A. SDS-PAGE analysis of one set of purifications. BL21 cells were grown and induced overnight to produce FPs from pS381_ara_FP vectors and purified as described in the text. Fractions: S (soluble proteins post lysis), F (flowthrough fraction – not bound by His column) and E (eluate from His column) were separated by SDS-PAGE and visualised by PAGEBlue staining (total protein stain). **B. Quantification of FP fraction by densitometry.** The bands in the S lanes (indicated with red asterisks) were quantified relative to the density of all bands in those same lanes using ImageJ. (The bands from the eluate lanes (indicated with black asterisks) were used to validate the choice of band in the lysate lanes.) The FP fraction (as percentage) was taken as FP density/Total density*100.

Finally, this phenomenon is also apparent in the yields obtained from our four sets of lysate preparations:

set	yield (ng/μl)		
	mCherry	mGFPmut3	mTagBFP2
1	50	225	77
2	52	142	84
3	32	89	53
4	59	102	79
mean	48	140	73

Rebuttal Table 2. Yields of protein in lysates. Protein concentrations in the first (most concentrated) wells of the corresponding serial dilutions from all sets of lysates used in Supplementary Fig. 11A-B, rounded to nearest ng/μl.

It is clear from the above that the abundance of FPs that accumulate in *E. coli* even using identical vectors that differ only in the FP coding sequence can vary by over 3-fold. In this case we observed an almost 2-fold difference between mCherry and mTagBFP2. The data in Fig. 5 shows a larger difference, but the pattern of higher mTagBFP2 abundance is maintained. The more pronounced difference in Fig. 5 may be down to the use of lower copy vectors, a different *E. coli* strain, different media and different mode of expression (lower volumes, in a 96-well plate, in plate reader). The broader point, that identical vectors that differ only in their FP coding sequences can produce varying amounts of protein, remains.

- The authors have still not provided any means of determining whether the protocol has been executed correctly. While I am quite willing to believe that the authors have executed correctly and with great care, the same will not be true for new users of the method whose understanding will be imperfect. This issue is particularly important because of the complexity of the method and the ample opportunities for opaque failures during its execution. What process controls can be used to determine whether a user of the method should trust the results that they have generated?

We thank the reviewer for this comment. We agree that protocols should be clear and, if possible, include ways of identifying obvious errors. The final protocol is in fact extremely simple, and as a result requires few experimental validations or controls. As a result, most validations are conducted analytically.

Initially, the preparation of the FP calibrants was more complex in the sense that it required numerous steps and did indeed require numerous validation steps and process controls described in the text and in Fig. 2, which were necessary for establishing the purified calibrants were of good quality. However, the final protocol is much simpler (Fig. 1, Fig. 3D). It consists of essentially 4 steps (1: expression, 2: calibrant preparation, 3: protein assay, and 4: fluorescence assay), and the calibrant preparation step has been reduced to only 3 sub steps (A: lysis, B: clarification, and C: concentration). The final protocol was initially published to protocols.io, and it has now been incorporated into the paper itself for clarity (Supplementary Note 5). If it appears long, that is because we have endeavoured to add enough detail such that new users will have no trouble reproducing any of the steps. Many of the original process controls are no longer necessary as purification is no longer required.

Nonetheless, users can bear in mind the following, which we have added to the end of the protocol in Supplementary Note 5:

“Validation steps and controls

Expression

- For some FPs (particularly bright greens and reds), it will be clear by eye if expression levels are high after overnight culture, as the culture will be brightly coloured. If this is not visible in the culture itself, it can become apparent after the first wash step as a brightly coloured cell pellet. For some of these proteins, the absence of colour typically means the expression conditions need optimisation. (Note that some FPs, such as mTagBFP2, will not produce a visible colour.)
- Overnight expression should produce high levels of protein but should not kill the cells: if using a plasmid that results in poor growth (less than 20 OD600 units of cells after overnight culture), use a lower inducer concentration, a lower growth temperature, or some other means to reduce cellular burden.
- Expression should result in adequate to decent yields of soluble protein and minimal aggregated protein. We do not see aggregation with our FPs, although in principle this can be a problem in protein overexpression that would limit FP yields. The presence of aggregates can be checked by SDS-PAGE after the clarification step that separates the soluble and insoluble fraction. The presence of a prominent FP-sized band in the insoluble fraction could indicate protein aggregation. This can often be resolved by using a lower inducer concentration or a lower growth temperature.
- The method has been validated with plasmids that are available on Addgene. These can be used as positive controls if users' own plasmids do not produce enough protein or produce unexpected results.

Calibrant preparation

- The lysed solution should go from turbid to clear. If it doesn't, this suggests the lysis was inefficient: repeat the lysis or increase the sonication amplitude or time.
- The clarification step that separates the soluble proteins from the insoluble proteins may be validated by separating the proteins of each fraction (supernatant: soluble proteins; pellet: insoluble proteins) by SDS-PAGE, as shown in Fig. 2C.
- Requirement for concentration: In principle, lysates may not always need to be concentrated prior to calibration, but in practice it is recommended to ensure that there is a high enough concentration in

the first few dilutions to get accurate protein concentration measurements from the absorbance readings. The total volume of lysate required may in some cases require trial and error, but 1.6ml (equivalent of 16 OD cells) is a good starting point. If 40 OD cells were lysed, up to 4ml lysate can be concentrated at this step. Another good rule of thumb (for green and red FPs at least) is that the FP stock should be concentrated enough to produce visibly colourful solutions.

Protein assay

- Typical raw data traces from absorbance spectra are provided in Supplementary Fig. 6A. Note the absorbance in the 900-1000nm range used for path length calculation, and the FP-specific absorbance in the intermediate wavelengths. Raw spectra that include wavelengths below 300nm will also contain a characteristic overflow absorbance in the UV range (most plastic microplates absorb in the UV range).
- If the peak is not evident by eye, this can indicate a low starting concentration of protein. After normalisation these usually become apparent, but if they do not, a higher starting concentration of protein is needed.

Fluorescence assay

- Visual inspection of raw fluorescence data can usually determine whether the dilutions are reasonably accurate and the replicates reasonably uniform.

“

In addition to this, the FPCountR R package includes a number of automated validations and checks, backed up by the ability of users to follow the progress of data processing steps via the production of plots and other files and to verify that it produces sensible results. For example, the function `get_conc_ECmax()`, that obtains concentration values for the calibrant dilution series from absorbance data, produces plots of the FP dilution:concentration data with its fit both without and with light scatter correction, as well as a CSV file with statistics of the linear regressions, to allow users to assess the quality of the fits and check that the correction is appropriate. The function `generate_cfs()` produces a CSV file that contains conversion factors calculated from the fit between FP concentration and RFU values from the pipetting model from Beal et al., 2020. In addition, it includes the residuals of that fit, allowing the assessment of validity as set out in that paper, which considered calibrations valid if the geometric mean of the absolute residuals was less than 1.1.

Typical and expected output plots and CSV files are illustrated in the Worked Example on the FPCountR website: <https://ec363.github.io/fpcountr/articles/fpcountr.html>. These can be recreated by users from the example data included with the FPCountR package.

- The authors continue to claim that their work should not be compared with dye-based methods, because dye molecules have a different spectrum. This overlooks the obvious option of computing a conversion from the excitation and emission spectra of the two molecules and the excitation and emission conditions of measurement. While the spectral conversion method is imperfect, so is FPCountR, and some of the variation that is now exposed in the revision makes me guess that the accuracy of FPCountR is likely on the order of 2-fold to 4-fold off from the true value, which I would guess may be less accurate than the spectral conversion method.

We thank the reviewer for this comment. Please see our response to the first comment, where we have included a fluorescein calibration.

Due to these shortcomings, at present I would advise against anyone making use of FPCountR, because:

- 1) I do not know whether its protein counts are close to correct or an order of magnitude high or low, and
- 2) there are no process controls that can help a person determine whether they have executed FPCountR correctly or have bungled it.

I do think there is a good chance that FPCountR is indeed a significant advance, and if this can be established, I would love to make use of it. Determining accuracy and providing process controls, however, are sine qua non for a calibration method.

There are some other issues that I believe still need to be addressed as well, but they are all much smaller and could be handled through minor revisions if the core concerns can be addressed.

We hope that the evidence we provide answers the points raised by the reviewer satisfactorily and gives further confidence in the validity and accuracy of our results.

Reviewer #2 (Remarks to the Author):

The authors have thoroughly responded to my feedback and I feel they have addressed the major comments in my initial review. My only minor comment is that the labels em_max and ex_max in Supplementary Table 7 are the wrong way around. I am happy to recommend this paper for publication.

We thank the reviewer for their comments, which we feel has helped this paper improve. We have amended the typo in Supplementary Table 7.

Reviewer #1 (Remarks to the Author):

Thank you to the authors for taking my concerns about accuracy and process controls seriously.

I particularly appreciate the use of spectral-translated comparison with fluorescein as an independent validation, and the inclusion of specific tests for detecting and debugging common failure modes.

I am happy to recommend acceptance for the manuscript in this form.

REVIEWERS' COMMENTS

Reviewer #1 (Remarks to the Author):

Thank you to the authors for taking my concerns about accuracy and process controls seriously.

I particularly appreciate the use of spectral-translated comparison with fluorescein as an independent validation, and the inclusion of specific tests for detecting and debugging common failure modes.

I am happy to recommend acceptance for the manuscript in this form.

We thank the reviewer for their thoughtful comments on the previous versions of the manuscript. We are happy that we were able to respond satisfactorily to all points and that their consideration has greatly strengthened our manuscript.